# Scalable Neural Video Representations
# with Learnable Positional Features

**Subin Kim**[*,1]     **Sihyun Yu**[*,1]     **Jaeho Lee**[2]     **Jinwoo Shin**[1]

[1]Korea Advanced Institute of Science and Technology (KAIST)

[2]Pohang University of Science and Technology (POSTECH)
{subin-kim, sihyun.yu, jinwoos}@kaist.ac.kr
jaeho.lee@postech.ac.kr

## Abstract

Succinct representation of complex signals using coordinate-based neural representations (CNRs) has seen great progress, and several recent efforts focus on extending them for handling videos. Here, the main challenge is how to (a) alleviate a compute-inefficiency in training CNRs to (b) achieve high-quality video encoding while (c) maintaining the parameter-efficiency. To meet all requirements (a), (b), and (c) simultaneously, we propose *neural video representations with learnable positional features* (NVP), a novel CNR by introducing "learnable positional features" that effectively amortize a video as latent codes. Specifically, we first present a CNR architecture based on designing 2D latent keyframes to learn the common video contents across each spatio-temporal axis, which dramatically improves all of those three requirements. Then, we propose to utilize existing powerful image and video codecs as a compute-/memory-efficient compression procedure of latent codes. We demonstrate the superiority of NVP on the popular UVG benchmark; compared with prior arts, NVP not only trains 2 times faster (less than 5 minutes) but also exceeds their encoding quality as 34.07→34.57 (measured with the PSNR metric), even using >8 times fewer parameters. We also show intriguing properties of NVP, *e.g.*, video inpainting, video frame interpolation, etc.[1]

## 1 Introduction

Recent advances in coordinate-based neural representations (CNRs) [9, 13, 17, 40, 46] have shown great promise in the field as a new paradigm for representing complex signals, including gigapixel images [29, 34], audios [40], 3D scenes [30, 33, 35], and even large city-scale street views [47]. Instead of storing signal outputs as a coordinate grid (*e.g.*, image pixels), whose memory requirement scales unfavorably in terms of resolution and dimension, CNRs represent each signal as a compactly parameterized, continuous neural network; they interpret a signal as a coordinate-to-value function and train a neural network to approximate this mapping. CNRs enjoy numerous appealing properties, including data compression [11, 12, 59], super-resolution [6], novel view synthesis [18, 22, 33, 54], and generative modeling of complex, high-dimensional data [14, 24, 41, 42, 57, 60], while being parameter-efficient interpretation of a given signal in various scenarios.

In particular, several works have attempted to exploit CNRs to interpret *video signals* [5, 23, 42, 57] by learning a neural network $f : \mathbb{R}^3 \to \mathbb{R}^3$ with $f(x, y, t) = (r, g, b)$ and exhibited their potential as a succinct representation of videos, as well as providing numerous applications including video generative modeling [42, 57], video compression [5, 59], and video super-resolution [7]. They

---

[*]Equal contribution.

[1]Videos are available at the website https://subin-kim-cv.github.io/NVP.

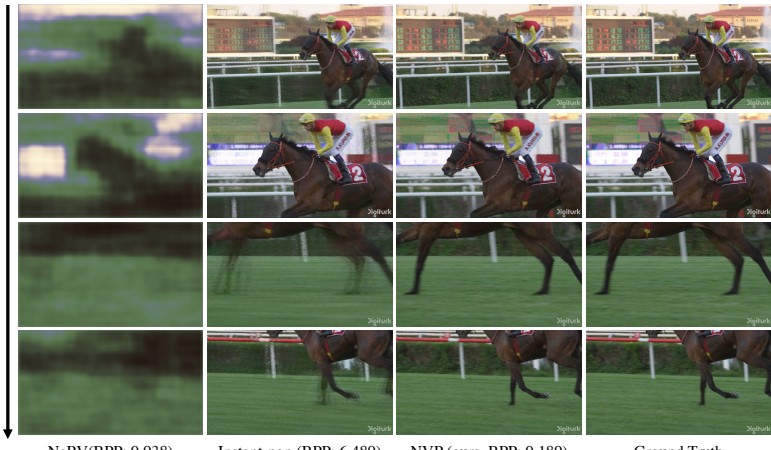

| NeRV(BPP: 0.938) | Instant-ngp (BPP: 6.489) | NVP (ours, BPP: 0.189) | Ground Truth |

Figure 1: Reconstruction results on Jockey in UVG-HD after training each model for "1 minute" with a single NVIDIA V100 32GB GPU. NVP can capture the detail of a video containing dynamic motions, *e.g.*, the legs of a running horse, while the prior methods generate blurry artifacts.

observe that conventional CNR architectures [40, 46] often fail to encode large-scale videos due to their complex temporal dynamics accompanied by large spatial variations, but the problem can be remarkably mitigated by designing CNR architectures specialized for videos. For instance, Chen et al. [5] proposes a CNR structure that focuses on the continuous modeling of the video signal only along the temporal dimension, allowing for more radical variations along spatial axes; it exhibits a comparable encoding quality to existing powerful video codecs (*e.g.*, H.264 [51], HEVC [43]) while enjoying lots of intriguing properties (*e.g.*, denoising and video frame interpolation).

However, CNRs suffer from a severe compute-inefficiency,[2] limiting their scalability to encode real-world, large-scale videos despite their advantages. To alleviate this issue, several works [26, 34, 37] have proposed new CNR architectures by separating a CNR $f$ into two parts; $f = h_\phi \circ g_\theta$ for a coordinate-to-latent mapping $g_\theta(x, y, t) = \mathbf{z}$ and a latent-to-RGB mapping $h_\phi(\mathbf{z}) = (r, g, b)$. They construct $g_\theta$ as an embedding function defined with *latent grids* $\mathbf{U}_\theta$ in which the shape resembles the grid interpretation of a given signal (*e.g.*, a 2D array of $C$-dimensional latent codes $\mathbf{U}_\theta \in \mathbb{R}^{H \times W \times C}$ for image pixels) rather than as a neural network. These approaches have shown a promise in compute-efficiency due to the strong locality induced by a grid structure of $\mathbf{U}_\theta$; however, they result in another problem: these architectures severely sacrifice the parameter-efficiency since the parameter size of $\theta$ can be very large, growing proportionally to both the input coordinate dimension and the signal resolution. In this paper, we focus on developing video CNRs that are the best of both worlds: achieving high-quality encoding and the compute-/parameter-efficiency simultaneously.

**Contribution.** We introduce *neural video representations with learnable positional features* (NVP), a novel CNR for videos. NVP avoids requiring a single giant full-dimensional 3D array in $g_\theta$ by presenting *learnable positional features* that effectively amortize a given video as "2D and 3D" latent grids with succinct parameters. Specifically, we decompose the coordinate-to-latent mapping as

$$g_\theta = \underbrace{g_{\theta_{xy}} \times g_{\theta_{xt}} \times g_{\theta_{yt}}}_{\text{2D keyframes}} \times \underbrace{g_{\theta_{xyt}}}_{\text{3D sparse features}}, \qquad \text{for} \quad \theta := (\theta_{xy}, \theta_{xt}, \theta_{yt}, \theta_{xyt}),$$

where we present two types of latent grids for constructing these mappings (see Figure 2).

○ *Latent keyframes*: We first design "image-like" 2D latent grids $\mathbf{U}_{\theta_{xy}}$, $\mathbf{U}_{\theta_{xt}}$, $\mathbf{U}_{\theta_{yt}}$ for $g_{\theta_{xy}}, g_{\theta_{xt}}, g_{\theta_{yt}}$ (respectively) that learn the representative video contents across *each* spatio-temporal axis and dramatically improve the parameter efficiency of NVP.[3]

○ *Sparse positional features*: We then introduce a "video-like" 3D latent grid $\mathbf{U}_{\theta_{xyt}}$ for $g_{\theta_{xyt}}$, whose size is much smaller than the original video pixels, but effectively encodes video details locally.

---

[2]Measured with a single NVIDIA V100 32GB GPU, NeRV [5] takes at least 15 GPU hours to encode a single video of 600 frames with a $1920 \times 1080$ resolution for the desired quality.

[3]Such a spatio-temporal consideration is different from conventional approaches for specifying keyframes deterministically across only the temporal direction.

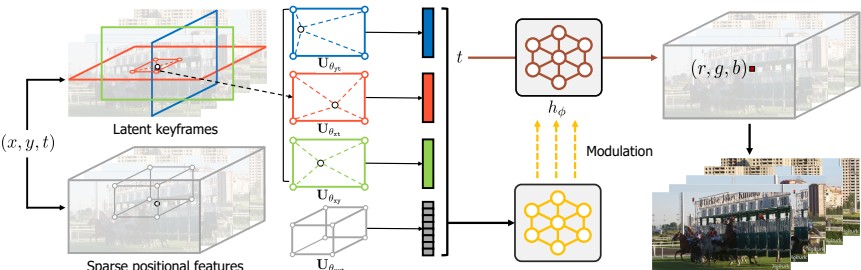

Figure 2: Overall illustration of our NVP. At a given space-time coordinate, NVP computes a latent vector from latent keyframes (see Figure 19 in Appendix E for details) and sparse positional features. The latent vector is passed through a neural network to compute the corresponding RGB output.

Moreover, we propose a compute-/memory-efficient compression procedure to further reduce the parameter $\theta$ by incorporating existing image and video codecs, *e.g.*, JPEG [49] for images and HEVC [43] for videos, respectively. In particular, we treat 2D and 3D latent grids like the image or video pixels and utilize powerful compression pipelines for them. Our compression scheme does not require any re-training of trained parameters, which significantly increases compute-efficiency to prior approaches to compressing CNR parameters but remarkably maintains the encoding quality. We also remark that such a compression approach is not applicable to the hashing-based latent grid of the prior method [34], while ours is "collision-free" and maintains the video- or image-like structures. Finally, for the choice of $h_\phi$, we suggest using a modulated network (with respect to the temporal coordinate) to improve the encoding quality of videos that contain dynamic motions.

We verify the effectiveness of our method on the popular UVG benchmark [32]. In particular, NVP achieves the peak signal-to-noise ratio (PSNR; higher is better) metric of 34.57 in 5 minutes (with a single NVIDIA V100 32GB GPU): it is achieved >2 times faster, even with using >8 times fewer parameters than the state-of-the-art on compute-efficiency that reaches 34.07 in 10 minutes. Moreover, compared with prior arts on encoding quality, our method improves the learned perceptual image patch similarity (LPIPS [58]; lower is better) as 0.145→0.102 (+29.7%) with a similar number of parameters while requiring ∼72.5% less training time. We also show numerous compelling properties of NVP, *e.g.*, video inpainting, video frame interpolation, super-resolution, compression, and consistent frame-wise encoding results without deviating quality.

## 2 Related work

**Coordinate-based neural representations.** Coordinate-based neural representations (CNRs), also known as implicit neural representations or neural fields, have emerged as a new paradigm for representing complex, continuous signals. They propose to encode signals through a neural network, typically a multilayer perceptron (MLP) combined with high-frequency sinusoidal activations [40] or Gaussian activations [9]. Prior works have focused on utilizing neural fields on various complicated data, *e.g.*, gigapixel images [29, 34], 2D videos [5, 42, 57], 3D static scenes, [30, 33, 35], and 3D dynamics scenes [23, 39, 52]. In particular, most approaches have focused on constructing CNR architectures for encoding 3D scenes [3, 15, 47] and exhibited how employing specialized prior knowledge for a given signal domain in the architecture can remarkably boost the encoding quality. Despite these successes, extending CNRs for videos is yet under-explored. In this paper, we aim to move toward developing a CNR for videos by exploiting their unique temporal properties.

**Hybrid CNRs.** Rather than solely designing a neural network of coordinate-to-RGB mapping, hybrid CNRs incorporate learnable latent codes that follow a grid structure, *e.g.*, image CNRs combined with 2D latent spatial grids [6, 31], and 3D scene (or shape) CNRs with latent cubic grids [4, 8, 19, 26, 29, 37]. Specifically, they compute a latent vector using the grid-structured latent code and pass it through a neural network to compute the signal output at a given input coordinate. Such approaches have shown significant efficiencies in training time and encoding quality due to the powerful locality induced by grid-shaped latent codes. However, the number of parameters required for latent grid-based representations easily grows proportionally to the input coordinate dimension or data resolution, limiting the scalability of hybrid CNRs. Remarkably, some of the recent approaches have exhibited this inefficiency can be significantly mitigated by considering multi-level (or progressive) structures for latent grids [26, 29, 44, 45]. While prior works primarily focus on encoding images or 3D scenes, we aim to design parameter-efficient hybrid CNRs for videos.

# 3 NVP: Neural video representations with learnable positional features

We first formulate our problem setup as follows. Given a video signal $\mathbf{v} := (\mathbf{f}_1, \mathbf{f}_2, \ldots, \mathbf{f}_T)$ consisting of $T$ video frames, the goal is to find a compact neural representation $f_{\mathbf{w}}$ with parameters $\mathbf{w}$, from which the original video $\mathbf{v}$ can be reconstructed with high quality. Here, the quality can be defined using various distortion metrics, *e.g.*, peak signal-to-noise ratio (PSNR) [16] and LPIPS [58], for evaluating a reconstruction quality and a perceptual similarity, respectively.

To achieve this goal, we take an approach based on *coordinate-based neural representations* (CNRs)— a paradigm where each datum (*e.g.*, video) is parameterized as a neural network of coordinate mapping. In particular, we aim to represent the given video using a neural network $f_{\mathbf{w}} : \mathbb{R}^3 \to \mathbb{R}^3$, which maps the space-time coordinates $(x, y, t)$ of the video to corresponding RGB values $(r, g, b)$, where $f_{\mathbf{w}}$ is optimized with reconstruction objectives, *e.g.*, mean-squared error. Such an approach has significant potential, as CNRs have shown to encode other continuous, complex signals (*e.g.*, 3D scenes) [40, 46] compactly while enjoying lots of intriguing properties, *e.g.*, super-resolution [6] and denoising [5]. However, CNRs have suffered from tremendous time costs for training, and even in the case of videos, it is difficult to achieve high-quality encodings if one utilizes conventional CNR architectures that overlook the complex spatio-temporal dynamics of videos [5]. Our contribution lies in resolving these issues by designing "learnable positional features" that succinctly encode a video as latent codes with high quality and keeping their compute-/parameter-efficiency intact.

In the rest of this section, we provide a detailed description of each component in NVP. In Section 3.1, we explain the architecture of NVP. We then describe our compression procedure in Section 3.2.

## 3.1 Architecture

We design our video CNR $f_{\mathbf{w}}$ as a composition of two functions with a parameterization $\mathbf{w} := (\theta, \phi)$: a coordinate-to-latent mapping $g_\theta$ and a latent-to-RGB mapping $h_\phi$. Here, we decompose the coordinate-to-latent-mapping as $g_\theta = g_{\theta_{\mathsf{xy}}} \times g_{\theta_{\mathsf{xt}}} \times g_{\theta_{\mathsf{yt}}} \times g_{\theta_{\mathsf{xyt}}}$ with $g_\theta(x, y, t) = (\mathbf{z}_{\mathsf{xy}}, \mathbf{z}_{\mathsf{xt}}, \mathbf{z}_{\mathsf{yt}}, \mathbf{z}_{\mathsf{xyt}})$ (for $\theta := (\theta_{\mathsf{xy}}, \theta_{\mathsf{xt}}, \theta_{\mathsf{yt}}, \theta_{\mathsf{xyt}})$), where each $g_{\theta_{\mathsf{xy}}}, g_{\theta_{\mathsf{xt}}}, g_{\theta_{\mathsf{yt}}}$ is formalized with image-like 2D latent spatial grids $\mathbf{U}_{\theta_{\mathsf{xy}}}, \mathbf{U}_{\theta_{\mathsf{xt}}}, \mathbf{U}_{\theta_{\mathsf{yt}}}$ (respectively) and $g_{\theta_{\mathsf{xyt}}}$ is designed with a video-like sparse 3D latent grid $\mathbf{U}_{\theta_{\mathsf{xyt}}}$. We then present the latent-to-RGB mapping $h_\phi(\mathbf{z}_{\mathsf{xy}}, \mathbf{z}_{\mathsf{xt}}, \mathbf{z}_{\mathsf{yt}}, \mathbf{z}_{\mathsf{xyt}}) = (r, g, b)$ to be a multi-layer perceptrion (MLP) modulated by another neural network. To explain our architecture, we assume all the input coordinate $(x, y, t)$ of $g_\theta$ (and $f_{\mathbf{w}}$) is in $[0, 1]^3 \subset \mathbb{R}^3$ without loss of generality.

**Learnable latent keyframes.** At a high level, learnable latent keyframes $\mathbf{U}_{\theta_{\mathsf{xy}}}, \mathbf{U}_{\theta_{\mathsf{xt}}}, \mathbf{U}_{\theta_{\mathsf{yt}}}$ are image-like 2D latent grids learned to capture the common representative contents in a given video $\mathbf{v}$ across each $t$-, $y$-, $x$-axis, respectively. For a given input coordinate $(x, y, t)$, we compute latent vectors $\mathbf{z}_{\mathsf{xy}}, \mathbf{z}_{\mathsf{xt}}, \mathbf{z}_{\mathsf{yt}}$ from $\mathbf{U}_{\theta_{\mathsf{xy}}}, \mathbf{U}_{\theta_{\mathsf{xt}}}, \mathbf{U}_{\theta_{\mathsf{yt}}}$ individually. We explain our keyframe only with $\mathbf{U}_{\theta_{\mathsf{yt}}}$ by letting $\mathbf{U} := \mathbf{U}_{\mathsf{yt}}$ for simplicity, but note that other keyframes $\mathbf{U}_{\theta_{\mathsf{xy}}}, \mathbf{U}_{\theta_{\mathsf{xt}}}$ operate in the similar manner.

Formally, $\mathbf{U}$ is $L$ 2D spatial grids of $C$-dimensional latent codes $u_{ij}$, whose resolution is $H_l \times W_l$:

$$\mathbf{U} := (U_1, \ldots, U_L),$$
$$U_l := (u_{ij}^l) \in \mathbb{R}^{H_l \times W_l \times C} \quad \text{for } l = 1, \ldots, L,$$
$$u_{ij} \in \mathbb{R}^C \qquad \text{for } 1 \leq i \leq H_l,\, 1 \leq j \leq W_l.$$

Here, the keyframe follows an $L$-level multi-resolution structure, *i.e.*, for each level $l$, the height $H_l$ and the width $W_l$ become different as $H_l = \lfloor \gamma^{l-1} H_1 \rfloor$ and $W_l = \lfloor \gamma^{l-1} W_1 \rfloor$ with fixed $\gamma > 1$ and $H_1, W_1 > 0$, where $\lfloor \cdot \rfloor$ indicates the floor function of the input. Since $\mathbf{U} = \mathbf{U}_{\theta_{yt}}$ is shared over the $x$-axis, we compute the latent vector $\mathbf{z}_{\mathsf{yt}} := (z_{\mathsf{yt}}^1, \ldots, z_{\mathsf{yt}}^L)$ by considering only the value of $y$ and $t$ at a given coordinate $(x, y, t)$. Specifically, for $l = 1, \ldots, L$, each $z_{\mathsf{yt}}^l$ is a linearly interpolated vector of four vectors in the spatial grid $U_l$, where the indices of these vectors are chosen as the closest ones to the relative position of the input coordinate $(y, t) \in [0, 1]^2$:

$$(m, n) = (\lfloor yH_l \rfloor, \lfloor tW_l \rfloor) \quad \text{for } l = 1, \ldots, L,$$
$$z_{\mathsf{yt}}^l = \texttt{lerp}\Big( (yH_l - m, tW_l - n); u_{mn}^l, u_{m,n+1}^l, u_{m+1,n}^l, u_{m+1,n+1}^l \Big),$$

where $\texttt{lerp}$ indicates a linear interpolation operation at the input coordinate between given vectors.

Note that we *learn* the keyframe as the latent codes, unlike conventional approaches that specify the keyframe in a deterministic manner from $T$ video frames $(\mathbf{f}_1, \mathbf{f}_2, \ldots, \mathbf{f}_T)$; it encourages to capture

the representative contents of the video over each spatio-temporal direction better. We also remark that our architecture involves two keyframes $\mathbf{U}_{\theta_{xt}}, \mathbf{U}_{\theta_{yt}}$ that are considered across spatial axes. Considering these keyframes may not be beneficial in the RGB space as the spatial variation of video pixels is often large. However, in our approach, these keyframes are learned under a more flexible, continuous latent space, and thus the representative frames can even be found in these spatial directions in such a space while encoding the RGB outputs of the video accurately.

Finally, recall that $\mathbf{U}_{\theta_{xy}}$, $\mathbf{U}_{\theta_{xt}}$, $\mathbf{U}_{\theta_{yt}}$ consist of multi-resolution spatial grids, *i.e.*, the resolution of spatial grids in each latent keyframe grows from coarse to fine. Since natural scenes often include repeating patterns in various scales, *e.g.*, a scene of flowers of different sizes, this multi-resolution architecture promotes learning common patterns with reduced memory and computation costs, which is also validated in Müller et al. [34].

**Sparse positional features.** Given a space-time coordinate $(x, y, t)$, we compute the latent vector $\mathbf{z}_{xyt}$ with $\mathbf{U}_{\theta_{xyt}}$ that represents the local details of the video at this input position. Here, $\mathbf{U}_{\theta_{xyt}} :=$ $(u_{ijk}) \in \mathbb{R}^{H \times W \times S \times D}$ is a 3D *sparse* grid of $D$-dimensional latent codes, *i.e.*, the 3D grid size $H \times W \times S$ is much smaller than the size of the video pixels of a 3D RGB grid:

$$\mathbf{U}_{\theta_{xyt}} := (u_{ijk}) \in \mathbb{R}^{H \times W \times S \times D}, \ u_{ijk} \in \mathbb{R}^D \quad \text{for } 1 \le i \le H, \ 1 \le j \le W, \ 1 \le k \le S.$$

To evaluate the latent vector $\mathbf{z}_{xyt}$, we concatenate $h \times w \times s$ latent codes in $\mathbf{U}_{\theta_{xyt}}$ that their indices are near the relative position of the input $(x, y, t)$:

$$(m, n, k) = (\lfloor xH \rfloor, \lfloor yW \rfloor, \lfloor tS \rfloor),$$
$$\mathbf{z}_{xyt} = (u_{mnk}, \dots, u_{m+h-1, n+w-1, k+s-1})$$

where $h, w, s > 0$ are given as hyperparameters.

The locality of $\mathbf{U}_{\theta_{xyt}}$ as 3D latent codes dramatically alleviates the compute-inefficiency in fitting videos, in contrast to conventional CNRs where the entire parameters are shared (as a neural network) for arbitrary input coordinates $(x, y, t)$ and thus require a significant training cost. Moreover, recall that we construct $\mathbf{U}_{\theta_{xyt}}$ as a sparse 3D grid of latent codes; remarkably, $\mathbf{U}_{\theta_{xyt}}$ efficiently captures the video details even if the size is smaller than the number of video pixels since the common contents of a given video are effectively encoded with the latent keyframes $\mathbf{U}_{\theta_{xy}}, \mathbf{U}_{\theta_{xt}}, \mathbf{U}_{\theta_{yt}}$.

Note that we design $\mathbf{U}_{\theta_{xyt}}$ as a sparse 3D grid; each single $u_{ijk}$ should represent a wide area of videos solely without concatenation. As the single latent vector may lack expressive power to represent such a wide range and may result in a non-smooth transition as the CNR output. Hence, we mitigate this issue by concatenating multiple vectors near each other (see Figure 10 in Section 4.3).

Instead of directly selecting near latent codes from $\mathbf{U}_{\theta_{xyt}}$, one may consider upsampling of $\mathbf{U}_{\theta_{xyt}}$ using linear interpolation before selecting the close latent codes at a given coordinate. Such an interpolation further helps each latent code to learn smoother representations and also generalizes to representing video frames at unseen coordinates during training. Meanwhile, this upsampling requires more computing time compared to the computational cost of other modules in NVP, and thus faces a trade-off between the smoothness and compute-efficiency in training (see Table 4 in Section 4.3).

**Modulated implicit function.** With the latent vector $\mathbf{z} := [\mathbf{z}_{xy}, \mathbf{z}_{xt}, \mathbf{z}_{yt}, \mathbf{z}_{xyt}]$ evaluated from $g_\theta$, a naïve design choice of the latent-to-RGB mapping $h_\phi$ is to utilize a MLP that maps $\mathbf{z}$ to the corresponding RGB output $(r, g, b)$. However, if a video contains temporally dynamic motions, we found such a simple MLP architecture occasionally lacks expressive power and can be difficult to capture the complex dynamics of the given video, even with the large network size of $h_\phi$.

To circumvent this issue, we design $h_\phi$ to be a $K$-layer MLP (coined as a synthesizer network) modulated by another modulator network [31]: where the latent vector $\mathbf{z}$ and the time coordinate $t$ are passed through the modulator and the synthesizer, respectively. Here, the modulator network utilizes piecewise linear activations (*e.g.*, ReLU), where the synthesizer uses sinusoidal activations. Specifically, an RGB output $(r, g, b)$ of the latent vector $\mathbf{z}$ (from $(x, y, t)$) is computed as follows:

$$\boldsymbol{\alpha}_0 = t,$$
$$\boldsymbol{\alpha}_k = \mathbf{z}_k \odot \sin(\boldsymbol{A}_k \boldsymbol{\alpha}_{k-1} + \boldsymbol{b}_k) \quad \text{for } k = 1, \dots, K-1,$$
$$(r, g, b) = \boldsymbol{A}_K \boldsymbol{\alpha}_{K-1} + \boldsymbol{b}_K,$$

where $\boldsymbol{A}_k, \boldsymbol{b}_k$ are weights and biases of $k$-th layer of the synthesizer, $\boldsymbol{z}_k$ is $k$-th hidden feature of the modulator, and $\odot$ denotes an element-wise product. It helps to achieve high-quality encoding rapidly in early training iterations and often at the convergence than a naive MLP (See Table 3 for details).

Table 1: PSNR, FLIP, and LPIPS of different CNRs to encode videos in UVG-HD under each encoding time. ↑ and ↓ denote higher and lower values are better, respectively. Subscripts denote standard deviations, and bolds indicate the best results. * indicates applying the method without the corresponding compression scheme. We report the BPP values of NeRV without compressing parameters if the encoding time is ≤ 1 hour since the NeRV's compression requires a longer time. On the other hand, the compression procedure of NVP only takes less than 1 minute, but for a fair comparison, we do not apply it to NVP as well whenever NeRV is not compressed.

| Encoding time | Method | BPP | PSNR (↑) | FLIP (↓) | LPIPS (↓) |
|---|---|---|---|---|---|
| ~5 minutes | Instant-ngp [34] | 7.580 | 33.15±3.19 | 0.090±0.034 | 0.226±0.112 |
| | NeRV-S* [5] | 1.072 | 24.16±5.17 | 0.219±0.097 | 0.542±0.180 |
| | **NVP-S* (ours)** | 0.901 | **34.57±2.62** | **0.075±0.021** | **0.190±0.100** |
| ~10 minutes | Instant-ngp [34] | 7.580 | 34.07±3.01 | 0.082±0.030 | 0.204±0.105 |
| | NeRV-S* [5] | 1.072 | 26.53±5.92 | 0.176±0.088 | 0.460±0.184 |
| | **NVP-S* (ours)** | 0.901 | **35.79±2.31** | **0.065±0.016** | **0.160±0.098** |
| ~1 hour | Instant-ngp [34] | 7.580 | 35.69±2.72 | 0.071±0.025 | 0.162±0.090 |
| | NeRV-S* [5] | 1.072 | 33.26±4.31 | 0.094±0.038 | 0.240±0.112 |
| | **NVP-S* (ours)** | 0.901 | **37.61±2.20** | **0.052±0.011** | **0.145±0.106** |
| ~15 hours | SIREN [40] | 0.284 | 27.20±3.77 | 0.169±0.059 | 0.409±0.124 |
| | FFN [46] | 0.284 | 28.18±3.62 | 0.153±0.055 | 0.442±0.126 |
| | Instant-ngp [34] | 0.229 | 28.81±3.48 | 0.155±0.057 | 0.390±0.135 |
| | NeRV-S [5] | 0.201 | 36.14±3.97 | **0.067±0.023** | 0.163±0.101 |
| ~8 hours | **NVP-S (ours)** | 0.210 | **36.46±2.18** | **0.067±0.017** | **0.135±0.083** |
| >40 hours | SIREN [40] | 0.284 | 26.09±3.88 | 0.175±0.082 | 0.486±0.188 |
| | FFN [46] | 0.284 | 29.53±3.44 | 0.135±0.052 | 0.391±0.124 |
| | Instant-ngp [34] | 0.436 | 29.98±3.39 | 0.138±0.051 | 0.358±0.140 |
| | NeRV-L [5] | 0.485 | 35.00±3.31 | 0.079±0.020 | 0.145±0.100 |
| ~11 hours | **NVP-L (ours)** | 0.412 | **37.47±2.08** | **0.062±0.017** | **0.102±0.061** |

## 3.2 Compression procedure

Recall that we aim to find "compact" video CNRs; several works have focused on reducing the number of coordinate-based neural representations parameters (or bits) after training while maintaining their performance. In particular, they have relied on exploiting existing well-known techniques for neural network compression, *e.g.*, exploiting magnitude pruning [5, 21] or quantization [5, 59], and exhibited considerable results. However, these approaches mainly involve a re-training of CNR parameters, requiring severe computation costs, and thus are not suitable for practical scenarios.

Instead, we propose a compression pipeline for NVP, which does not require re-training, yet significantly reduces the number of bits while preserving the video quality. Our main idea is to incorporate existing image and video codecs that have shown their promises for the compression of given pixels. In particular, we focus on compressing keyframes $\mathbf{U}_{\theta_{xy}}$, $\mathbf{U}_{\theta_{xt}}$, $\mathbf{U}_{\theta_{yt}}$, and sparse positional features $\mathbf{U}_{\theta_{xyt}}$, as the parameter size of the modulated implicit function $h_\phi$ is neglectable compared with those. Specifically, we quantize $\mathbf{U}_{\theta_{xyt}}$ and $\mathbf{U}_{\theta_{xy}}$, $\mathbf{U}_{\theta_{xt}}$, $\mathbf{U}_{\theta_{yt}}$ as 3D/2D grids of 8-bit latent codes and regard them as video and image pixel grids, where the number of the channel becomes the dimension of latent codes. Based on these interpretations, we compress these latent codes by utilizing existing video and image codecs, *e.g.*, HEVC [43] for videos and JPEG [49] for images. Intriguingly, we found this procedure can significantly reduce the parameters while notably maintaining the video quality without any fine-tuning of the latent-to-RGB mapping $h_\phi$ (See Section 4.3).

# 4 Experiments

We verify the effectiveness of our framework on UVG-HD [32], a representative benchmark for evaluating video encodings. Experimental results demonstrate that our neural video representations with learnable positional features (NVP) simultaneously improves the overall performance by (a) alleviating a compute-inefficiency in training, (b) achieving high-quality video encoding, and (c) maintaining parameter-efficiency. We also show applications of our NVP, including video inpainting and spatio-temporal interpolation. Finally, we conduct ablation studies to validate each component.

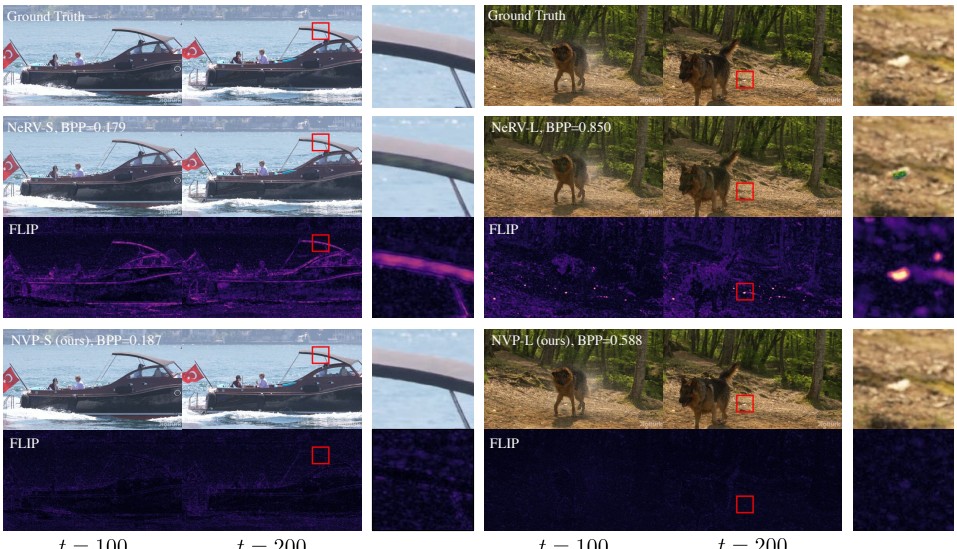

$t = 100$      $t = 200$           $t = 100$      $t = 200$

Figure 3: Illustration of reconstructions on Yachtride (left), and ShakeNDry (right) in UVG-HD. FLIP indicates the output of evaluating its metric. The red box is zoomed in as the image at the right.

**Evaluation.** We follow similar setups on prior work [5] proposing coordinate-based neural representations (CNRs) for videos. All reported numbers are averaged over 7 videos in UVG-HD: Beauty, Bosphorus, HoneyBee, Jockey, ReadySetGo, ShakeNDry, and Yachtride, along with the standard deviations unless otherwise specified. We also use the Big Buck Bunny video, which is used in NeRV [5]. For quantitative evaluation, we use the following metrics: peak signal-to-noise ratio (PSNR) for reconstruction quality, LPIPS [58], FLIP [2], and SSIM [50] for perceptual similarity, where all metrics are evaluated in a frame-wise manner and averaged over the whole video. We evaluate these metrics on different bits-per-pixel (BPP; lower is better) to evaluate the parameter efficiency; see Appendix A.1 for more description of the evaluation.

**Implementation details.** All main experiments, including baselines, are processed with a single GPU (NVIDIA V100 32GB) and 28 instances from a virtual CPU (Intel® Xeon® Platinum 8168 CPU @ 2.70GHz), where it takes at most ∼11 hours to run our method and ∼2 days to run other baselines. Moreover, we denote NVP-S and NVP-L as the model with latent code dimensions of sparse positional features to be 2 and 4, respectively; see Appendix A.2 for more details.

**Baselines.** We compare our method with SIREN [40], and FFN [46], which are well-known signal-agnostic CNR architectures, Instant-ngp [34] for state-of-the-art CNRs on compute-efficiency, and NeRV [5], which is a CNR specialized for videos. For all of the baseline methods, we follow their reported experimental setups. In particular, for NeRV [5], we use two configurations provided in the official implementation: NeRV-S and NeRV-L for a small and a large model, respectively. See Appendix B for a detailed description of baseline methods.

### 4.1 Main results

Figure 1, Figure 3, and Table 1 summarize quantitative and qualitative results of NVP and baselines. Remarkably, as shown in Figure 1, NVP can accurately capture dynamic motions and high-frequency details of a video, *e.g.*, the legs of a running horse, while prior state-of-the-art CNRs fail to achieve and show a blurry artifact. We also emphasize such a high-quality encoding is accomplished in "less than 1 minute", which supports the superior compute-efficiency of NVP in training.

Moreover, Table 1 verifies the effectiveness of NVP with quantitative evaluations, which outperforms all other baselines at varying encoding times from ∼5 minutes to >40 hours. In particular, NVP significantly improves LPIPS compared with previous methods, demonstrating how the encoded videos are perceptually similar to ground-truth videos. Such a result is also confirmed in Figure 3; NeRV shows a distortion that some pixels significantly deviate from the ground-truth outputs, while our method does not suffer from such artifacts. We also note that the variance of NVP is relatively small compared with other baselines, which shows the robustness of videos with diverse scenes and motion. See Appendix C and D for video-wise results and discussion on decoding time, respectively.

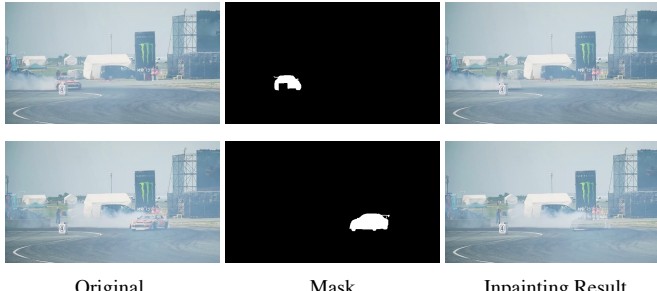

| | Original | Mask | Inpainting Result |

Figure 4: Video inpainting result of NVP on the drift-chicane video in DAVIS 2017 [38] to remove the masked car.

Table 2: Quantitative interpolation result of different methods on Big Buck Bunny measured with PSNR, LPIPS, and SSIM metrics. Bold indicates the best result.

| Metric | NeRV [5] | NVP (ours) |
|---|---|---|
| PSNR (↑) | 23.05 | **33.76** |
| LPIPS (↓) | 0.480 | **0.311** |
| SSIM (↑) | 0.690 | **0.960** |

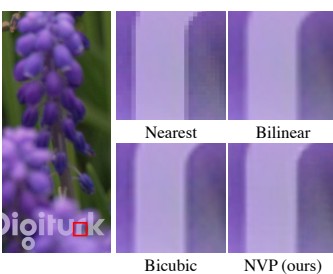

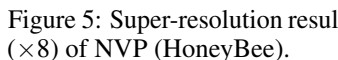

Figure 5: Super-resolution result (×8) of NVP (HoneyBee).

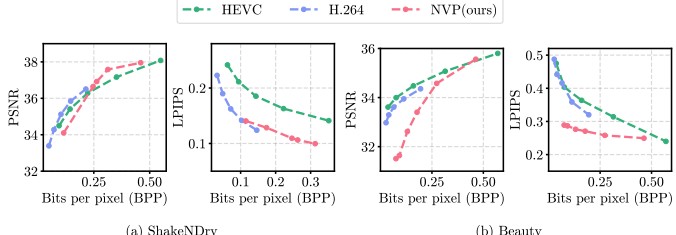

Figure 6: PSNR and LPIPS values of NVP and well-known video codecs over different BPP values computed on (a) the ShakeNDry video and (b) the Beauty video in UVG-HD.

## 4.2 Applications

In this section, we provide several applications of our method, NVP, as video CNRs. For better, playable illustrations and qualitative results, please refer to our project page.

**Video inpainting.** Intriguingly, our method has the capability of video inpainting, *i.e.*, the desired moving object in the video can be naturally removed by capturing shared video contents with learnable keyframes. Figure 4 visualizes the illustration of inpainting results from NVP; as shown in this figure, one can see the car is removed where such parts are filled with a natural background.

**Video frame interpolation.** Since video CNRs approximate videos as temporally continuous signals, they should interpolate among two different frames at an arbitrary time, even if such a frame does not exist in the training dataset. To validate the interpolation capability of NVP, we provide the quantitative results on Big Buck Bunny sequences; our method shows better interpolation results measured with various metrics, such as PSNR and LPIPS.

**Video super-resolution.** We remark that NVP encodes a given video as a *spatio-temporally* continuous signal. Thus, our method can interpolate the frames across spatial directions, *i.e.*, frame-wise super-resolution. Figure 5 exhibits how well NVP smoothly interpolates video frames across spatial directions while preserving the sharp edges, compared with naïve upsampling methods.

**Video compression.** Recall that one of the major advantages of CNRs is their succinct encoding to represent a given signal; one may consider utilizing CNRs for video compression [5, 11, 12]. To verify the potential of our method on video compression, we compare the quality of compressed videos from NVP with the ones from the current state-of-the-art video codecs. As shown in Figure 6, compressed videos from NVP show the comparable reconstruction quality (measured with PSNR metrics) while outperforming perceptual similarity (measured with LPIPS metrics).

## 4.3 Ablation studies

**Effect of architecture components.** To verify the effectiveness of each component, we train our model with all the videos in UVG-HD by removing each component while maintaining the total number of parameters, then measure PSNR metrics from these models. Table 3 summarizes the effect of three different architecture components. Without any of the components consisting of our positional features, the reconstruction quality gets dramatically worse, which validates how NVP

Table 3: PSNR values of each component of NVP: learnable keyframes, sparse feature, and modulation at 1,500 (1.5K) and 150,000 (150K) iterations. Bold indicates the scores within one standard deviation from the highest average score.

| Keyframes | Sparse feat. | Module. | # Params. | 1.5K | 150K |
|---|---|---|---|---|---|
| ✗ | ✓ | ✓ | 136M | $29.95_{\pm 2.69}$ | $31.21_{\pm 2.80}$ |
| ✓ | ✗ | ✓ | 138M | $29.88_{\pm 4.99}$ | $32.44_{\pm 4.48}$ |
| ✓ | ✓ | ✗ | 147M | $32.15_{\pm 3.08}$ | $\mathbf{38.04_{\pm 2.27}}$ |
| ✓ | ✓ | ✓ | 136M | $\mathbf{34.85_{\pm 2.69}}$ | $\mathbf{38.89_{\pm 2.11}}$ |

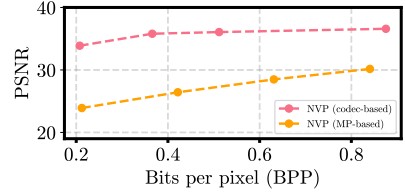

Figure 7: Rate-distortion plot of different compression strategies on ReadySetGo.

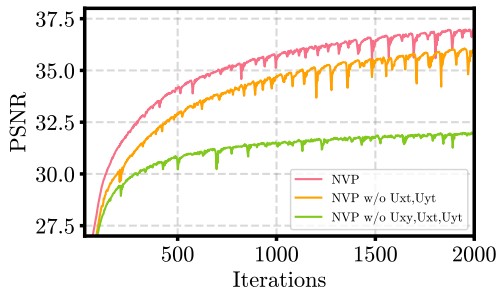

Figure 8: Convergence plot of NVP under different keyframe choices on the Jockey video.

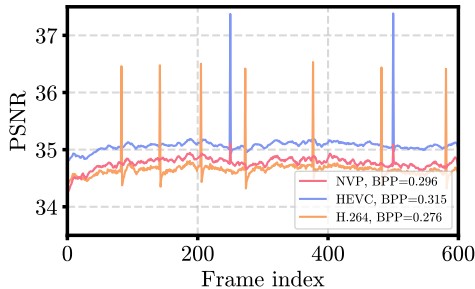

Figure 9: Frame-wise encoding quality of NVP and existing video codecs on Beauty.

succinctly encodes a given video as latent codes. We also note that the modulation not only improves the final encoding quality but also notably achieves high-quality encoding in its early training epochs.

**Compression procedure.** To validate the effectiveness of our compression scheme, we compare the encoding quality of (a) NVP with our compression scheme and (b) NVP compressed through magnitude-based pruning. Figure 7 shows the proposed compression pipeline outperforms conventional magnitude-based pruning under various BPP values. We also remark that our compression method does not require re-training and is thus much more time-efficient.

**Analysis of non-temporal keyframes.** Recall that we additionally design keyframes across spatial directions, unlike conventional approaches that designate the keyframe over only the temporal direction. To validate the effect of such keyframes, we compare NVP with (a) NVP without spatial keyframes and (b) NVP without all of the keyframes in Figure 8, where the number of parameters is equally set for a fair comparison. While utilizing latent learnable keyframes only over temporal direction is already fairly effective for high-quality encoding, one can observe the consideration of keyframes across other directions provides a further improvement.

**Consistent frame-wise encoding quality.** Existing keyframe-based compression approaches often suffer from inconsistent encoding quality: the frame-wise quality of the compressed video highly depends on whether it is the designated keyframe or not. In contrast, NVP *learns* the keyframes and does not have this problem. Figure 9 validates the result: our method exhibits consistent encoding performance while conventional popular video codecs show several peaks that the reconstruction quality (measured with PSNR metrics) highly deviates from others.

**Effect of the concatenation of latent codes in sparse positional features.** To validate the effectiveness of design choice on extracting latent features from sparse positional features $\mathbf{U}_{\theta_{xyt}}$, we compare the reconstructions from NVP with and without concatenation of $\mathbf{U}_{\theta_{xyt}}$. As shown in Figure 10, the concatenation of latent codes $u_{ijk}$ in $\mathbf{U}_{\theta_{xyt}}$ indeed mitigates non-smooth transitions between latent codes and captures sharp details in a given video better. In particular, without the concatenation, it results in undesirable artifacts (*e.g.*, showing discontinuous borders), validating our concatenation scheme for constructing latent representations from sparse positional features.

**Effect of upsampling of sparse positional features.** We also examine the effect of upsampling of the sparse positional features $\mathbf{U}_{\theta_{xyt}}$. Figure 11 shows the result: linear interpolation of $\mathbf{U}_{\theta_{xyt}}$ exhibits more smooth patterns for unseen coordinates during training. Meanwhile, we note that the upsampling requires 1.61 times more training time per iteration due to the additional computation bottleneck (see Table 4); however, regardless of upsampling, we remark that NVP still achieves notable compute-efficiency compared with prior state-of-the-art methods (such as Chen et al. [5]).

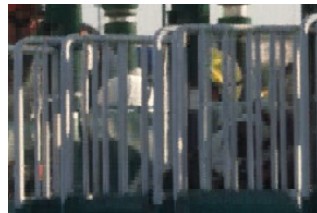 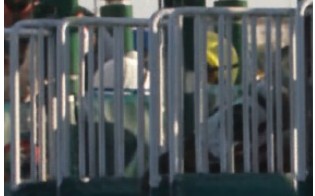 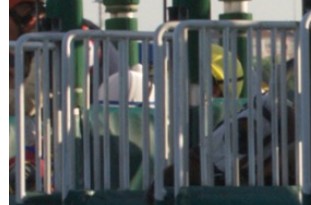

Without Concatenation of $\mathbf{U}_{\theta_{xyt}}$     With Concatenation of $\mathbf{U}_{\theta_{xyt}}$     Ground Truth

Figure 10: Reconstruction results on ReadySetGo in UVG-HD. Concatenation of sparse positional features captures sharp details (*e.g.*, a fence) better.

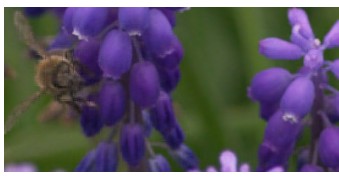 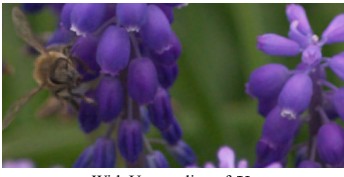

Without Upsampling of $\mathbf{U}_{\theta_{xyt}}$     With Upsampling of $\mathbf{U}_{\theta_{xyt}}$

Figure 11: Comparison of super-resolution result ($\times 8$) on HoneyBee.

Table 4: Training time per iteration with/without upsampling of sparse positional features.

| Upsampling | Time |
|:---:|:---:|
| ✗ | 0.291s |
| ✓ | 0.469s |

## 5  Discussion and conclusion

We proposed NVP, a new coordinate-based neural representation (CNR) to encode videos as succinct latent codes. Our main idea is to decompose a video into "image-like" and "video-like" structures to learn coordinate-to-latent mapping efficiently. Extensive experiments have verified the effectiveness of NVP on all the parameter-/compute-efficiency and the encoding quality. We hope our method will facilitate various future research directions in the CNR area.

**Limitations and future works.** Each video contains different scenes and motions so that it can be either static or dynamic, yet we utilize the same hyperparameters and architectures for encoding any video. Although such a video-agnostic design is fairly effective and outperforms prior works, we believe the video-wise consideration of the architecture and hyperparameter can remarkably boost the performance further. Moreover, we have shown the potential of utilizing powerful image and video codes for compressing latent codes in NVP; extending such codecs to be specialized for the compression of latent codes should be an interesting direction.

**Negative social impacts.** A side effect of CNRs is their potential unexpected behavior on encoding; they may cause undesirable artifacts in representing the given signal but are challenging to predict due to the under-explored behavior of training CNRs. Furthermore, in the case of representing videos, the encoded videos may suffer from severe distortions and conceivably cause ethical problems. In this respect, such behaviors should be extensively and carefully investigated and mitigated to exploit CNRs as the standard for encoding videos in real-world situations.

## Acknowledgments and Disclosure of Funding

We would like to thank Younggyo Seo, Jihoon Tack, Jongheon Jeong, Sukmin Yun, Jongjin Park, Junsu Kim, Seong Hyeon Park, Seojin Kim, Changyeon Kim, Jaehyun Nam, and anonymous reviewers for their helpful feedbacks and discussions. We also appreciate Max Ehrlich for providing the exact results of prior video compression methods.

This work was mainly supported by Institute of Information & communications Technology Planning & Evaluation (IITP) grant funded by the Korea government (MSIT) (No.2021-0-02068, Artificial Intelligence Innovation Hub; No.2019-0-00075, Artificial Intelligence Graduate School Program (KAIST); No.2019-0-01906, Artificial Intelligence Graduate School Program (POSTECH)). This research was partly supported by the Challengeable Future Defense Technology Research and Development Program (No.915027201) of Agency for Defense Development in 2022. This work was partially supported by the National Research Foundation of Korea (NRF) grant funded by the Korea government (MSIT) (No.2022R1F1A1075067).

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
