# Appendix: Scalable Neural Video Representations with Learnable Positional Features

**Website:** https://subin-kim-cv.github.io/NVP

## A  More description of experimental setups

### A.1  Metrics

- **PSNR**. To measure the reconstruction quality of models, we use peak signal-to-noise ratio (PSNR), evaluated as $-\log_{10}(\texttt{MSE})$, where $\texttt{MSE}$ denotes the mean-squared error between the ground-truth video and the reconstructed video (as 0-1 scale).

- **LPIPS**. We use AlexNet [20] pretrained on ImageNet [10], which best performs as a forward metric.[4] It measures a weighted distance between normalized internal features of the real video and its reconstruction.

- **FLIP**. FLIP automates the difference evaluation between alternating images by building on principles of human perception, where we use the official implementation.[5]

### A.2  Further implementation details

**Training details.** We train the network by adopting mean-squared error as our loss function and using the AdamW optimizer [27] with a learning rate of 0.01. We utilize HEVC [43] for the compression of our sparse positional features and JPEG [49] for reducing the parameters of keyframes.

**Architecture.** For the learnable keyframes, which follow a multi-level structure, we set the number of levels as 16, per level scale $\gamma$ as 1.35, and the coarsest resolution as $16 \times 16$. To build models with different sizes, we set the number of features per level as 2 and 4 for NVP-S and NVP-L, respectively. For the *sparse* positional features, considering the number of video frames, we use a 3D grid size of $300 \times 300 \times 300$ for ShakeNDry and $300 \times 300 \times 600$ for other videos in UVG-HD, which is $\sim 23\times$ smaller than the video pixels. We then concatenate $3 \times 3 \times 1$ latent codes in sparse positional features. Same as the learnable keyframes, we set the latent dimensions of sparse positional features to be 2 and 4 for NVP-S and NVP-L, respectively. In addition, we design modulated implicit function as a 3-layer multi-layer perceptron (MLP) modulated by another modulator network; both have a hidden size of 128. For the synthesizer network, we use SIREN [40], which uses $\sin(\sigma_t \mathbf{z})$ with $\sigma_t \in \mathbb{R}$ as the activation functions, and set the temporal frequency of the first layer as $\sigma_t = 30$ and the other layers as $\sigma_t = 1$. This network is modulated by a modulator network with LeakyReLU [53] activation.

We train the network with a batch size of 1,245,184, *i.e.*, at each iteration, we randomly sample 1,245,184 pixels from entire video pixels. We initialize the parameters of learnable keyframes (and sparse positional features) from the uniform distribution $U(-10^{-4}, 10^{-4})$ and use Kaiming normal initialization for the modulator networks. We train the network using AdamW optimizer [27] with the initial learning rate $\eta = 0.01$ and weight decay $\lambda = 0.001$. We use a cosine annealing learning rate scheduler, where the current learning rate $\eta_t$ at the $t$-th iteration is defined as follows:

$$\eta_t = \eta_{\min} + \frac{1}{2}(\eta - \eta_{\min})\Big(1 + \cos\Big(\frac{t}{T}\pi\Big)\Big),$$

where $\eta_{\min}$ is set to 0.00001, and the total iteration $T$ is set to 100,000 for both NVP-S and NVP-L.

**Compression procedure.** Following the prior work[5], we used *ffmpeg* [48] to extract RGB frames from compressed UVG-HD video files and to compress learnable positional features of NVP.

First, we use the following command to extract RGB videos from the original YUV videos of UVG-HD:

```
$ffmpeg -f rawvideo -vcodec rawvideo -s 3840x2160 -r 120 -pix_fmt yuv420p \
-i INPUT.yuv OUTPUT/f%05d.png
```

where `INPUT` is the input file name, and `OUTPUT` is a directory to save decompressed RGB frames.

---

[4] https://github.com/richzhang/PerceptualSimilarity
[5] https://github.com/NVlabs/flip

Then we use the following commands to compress learnable keyframes:

```
$ffmpeg -hide_banner -i input.png -qscale:v SCALE output.jpg
```

where `SCALE` is an output option that controls image quality.

We also use the following commands to compress sparse positional features:

```
$ffmpeg  -framerate FR -i INPUT/f%05d.png -c:v hevc -preset slow -x265-params \
bframes=0 -crf CRF OUTPUT.mp4
```

where `FR` and `CRF` indicate the frame rate and constant rate factor value that controls video quality and compression ratio, respectively. We summarize the hyperparameters used in the below table:

Table 5: Hyperparameter values used in the compression procedure.

| Model | SCALE $\mathbf{U}_{\theta_{xy}}$ | SCALE $\mathbf{U}_{\theta_{xt}}$ | SCALE $\mathbf{U}_{\theta_{yt}}$ | FR | CRF |
|---|---|---|---|---|---|
| NVP-S | 2 | 3 | 3 | 25 | 21 |
| NVP-L | 2 | 2 | 2 | 40 | 21 |

# B  Description of baseline methods

In this section, we briefly describe the specific parameter we used as baselines of video coordinate-based neural representations (CNRs) for evaluating our framework at a high level. To compare the parameter efficiency, we consider two situations: the average bits-per-pixel (BPP) of each baseline is either near 0.200 or 0.400. However, in the case of Instant-ngp [34], to compare the compute efficiency, we set the total number of parameters as similar as NVP-L when encoding time is $\leq 1$ hour

- **SIREN** [40], which uses high frequency *sine* activations (*i.e.*, $\sin(\omega_0 \mathbf{z})$ with $\omega_0 \gg 1$) and takes spatio-temporal coordinate $(x, y, t)$ as input then outputs a corresponding RGB value for each pixel. We use a 5-layer multi-layer perceptron (MLP) with a hidden size of 2,048, where the frequency $\omega_0$ is set to 30 for all sinusoidal activations.

- **FFN** [46] leverages random fourier feature (RFF) (*i.e.*, $[\sin(2\pi\mathbf{Wz}), \cos(2\pi\mathbf{Wz})]$) as a positional embedding layer to encode spatio-temporal coordinates $(x, y, t)$ and uses ReLU activation for further layers to output a corresponding RGB value for each pixel. We use RFF with $\mathbf{W} \in \mathbb{R}^{3 \times 1024}$ with $W_{ij} \sim \mathcal{N}(0, \sigma^2)$ with $\sigma = 10$, and a 4-layer MLP with a hidden size of 2,048.

- **NeRV** [5], a CNR specialized for videos, takes a time index as input and outputs a corresponding RGB image. We use two configurations provided in the official implementation:[6] NeRV-S and NeRV-L for a small and a large model, respectively. Specifically, we first apply a 2-layer MLP on the output of the positional encoding layer, and then we stack 5 NeRV blocks with upscale factors 5, 3, 2, 2, 2, respectively. We set the output channel for the first fully-connected layer as 128 for both models and change the expansion at the beginning of convolutional block size for 4 and 8 for NeRV-S and NeRV-L, respectively. After training one model for a single video separately, we prune the trained parameters by removing 15% of the weights, quantize model weights to 8-bit, and apply entropy coding following the compression procedure in the paper.

- **Instant-ngp** [34] uses multiresolution hash tables of trainable feature vectors for a given input coordinate $(x, y, t)$. To be specific, on the UVG-HD benchmark, we set the number of levels as 15, the number of features per level as 2, the maximum entries per level as $2^{24}$, and the coarsest resolution as 16. We set per level scale $\gamma$ as 1.3 for ShakeNDry, and 1.5 for other videos in the UVG-HD. We use a small neural network with 64 neurons and two hidden layers that use ReLU as output activation for a small latent-to-RGB mapping.

---

[6]https://github.com/haochen-rye/NeRV

# C  Video-wise main results

In this section, we provide video-wise quantitative results and qualitative illustrations.

Table 6: Quantitative results of NVP-S on 7 videos in UVG-HD: Beauty, Bosphorus, HoneyBee, Jockey, ReadySetGo, ShakeNDry, and Yachtride.

| Encoding time | Video name | BPP | PSNR ($\uparrow$) | FLIP ($\downarrow$) | LPIPS ($\downarrow$) |
|---|---|---|---|---|---|
| ~5 minutes | Beauty | 0.875 | 33.80 | 0.061 | 0.399 |
| | Bosphorus | 0.875 | 35.45 | 0.077 | 0.117 |
| | HoneyBee | 0.875 | 37.89 | 0.048 | 0.127 |
| | Jockey | 0.875 | 35.51 | 0.082 | 0.235 |
| | ReadySetGo | 0.875 | 30.74 | 0.109 | 0.161 |
| | ShakeNDry | 1.056 | 36.84 | 0.056 | 0.147 |
| | Yachtride | 0.875 | 31.73 | 0.090 | 0.146 |
| ~10 minutes | Beauty | 0.875 | 34.52 | 0.055 | 0.362 |
| | Bosphorus | 0.875 | 37.52 | 0.060 | 0.085 |
| | HoneyBee | 0.875 | 38.51 | 0.044 | 0.124 |
| | Jockey | 0.875 | 37.14 | 0.065 | 0.216 |
| | ReadySetGo | 0.875 | 32.37 | 0.094 | 0.112 |
| | ShakeNDry | 1.056 | 36.92 | 0.067 | 0.128 |
| | Yachtride | 0.875 | 33.53 | 0.074 | 0.096 |
| ~1 hour | Beauty | 0.875 | 35.42 | 0.048 | 0.361 |
| | Bosphorus | 0.875 | 40.06 | 0.048 | 0.065 |
| | HoneyBee | 0.875 | 39.51 | 0.039 | 0.121 |
| | Jockey | 0.875 | 38.91 | 0.050 | 0.201 |
| | ReadySetGo | 0.875 | 34.68 | 0.074 | 0.080 |
| | ShakeNDry | 1.056 | 38.81 | 0.045 | 0.118 |
| | Yachtride | 0.875 | 35.88 | 0.060 | 0.068 |
| ~8 hours | Beauty | 0.277 | 34.82 | 0.054 | 0.286 |
| | Bosphorus | 0.172 | 39.16 | 0.058 | 0.069 |
| | HoneyBee | 0.192 | 38.38 | 0.049 | 0.095 |
| | Jockey | 0.172 | 37.03 | 0.073 | 0.220 |
| | ReadySetGo | 0.181 | 33.49 | 0.094 | 0.090 |
| | ShakeNDry | 0.297 | 37.85 | 0.055 | 0.104 |
| | Yachtride | 0.180 | 34.49 | 0.083 | 0.084 |
| ~11 hours | Beauty | 0.523 | 35.43 | 0.051 | 0.170 |
| | Bosphorus | 0.333 | 40.21 | 0.056 | 0.049 |
| | HoneyBee | 0.409 | 39.13 | 0.044 | 0.064 |
| | Jockey | 0.323 | 37.86 | 0.067 | 0.205 |
| | ReadySetGo | 0.351 | 34.49 | 0.091 | 0.073 |
| | ShakeNDry | 0.585 | 38.70 | 0.052 | 0.088 |
| | Yachtride | 0.359 | 36.44 | 0.075 | 0.066 |

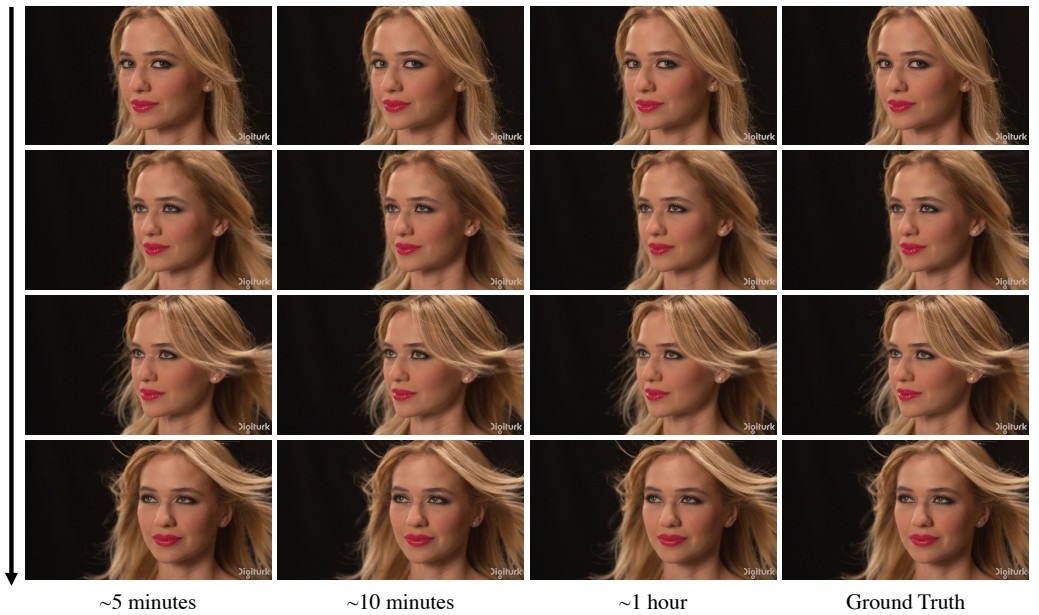

| ~5 minutes | ~10 minutes | ~1 hour | Ground Truth |

Figure 12: Reconstruction results on Beauty in UVG-HD after training NVP-S for "5 minutes", "10 minutes", and "1 hour" with a single NVIDIA V100 32GB GPU.

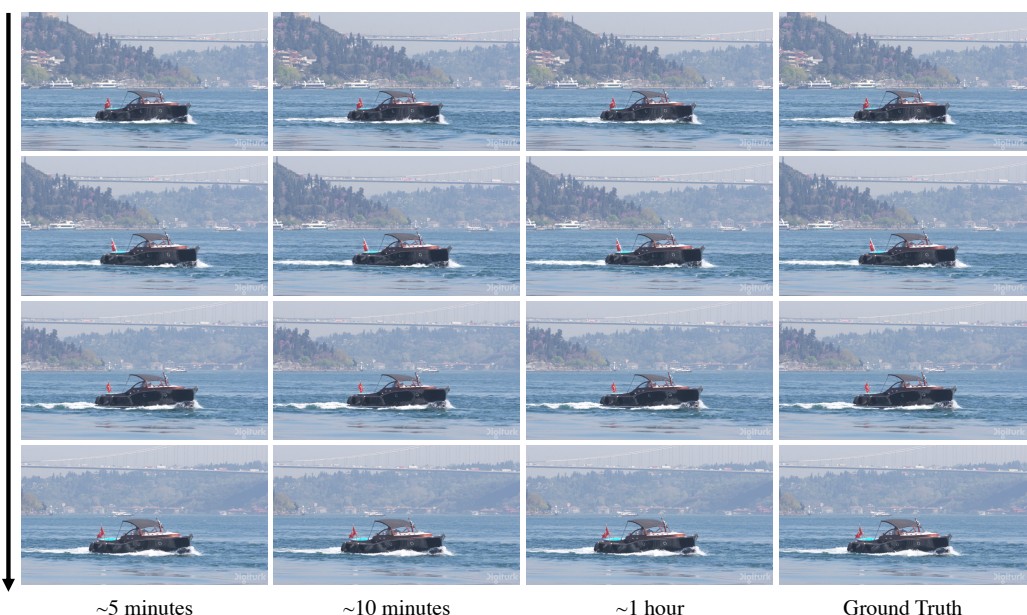

| ~5 minutes | ~10 minutes | ~1 hour | Ground Truth |

Figure 13: Reconstruction results on Bosphorus in UVG-HD after training NVP-S for "5 minutes", "10 minutes", and "1 hour" with a single NVIDIA V100 32GB GPU.

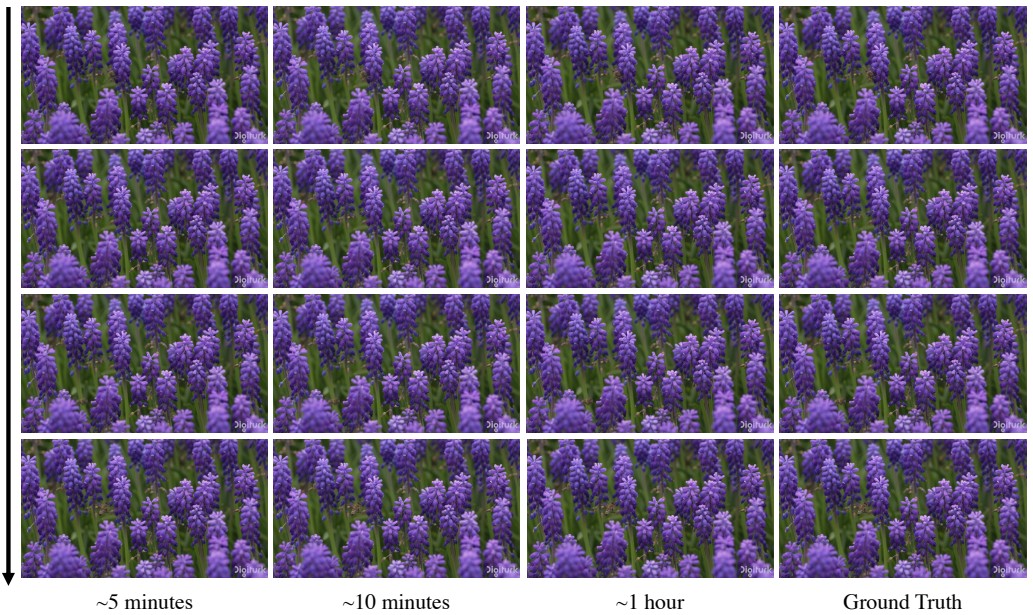

|  ~5 minutes | ~10 minutes | ~1 hour | Ground Truth |

Figure 14: Reconstruction results on Honeybee in UVG-HD after training NVP-S for "5 minutes", "10 minutes", and "1 hour" with a single NVIDIA V100 32GB GPU.

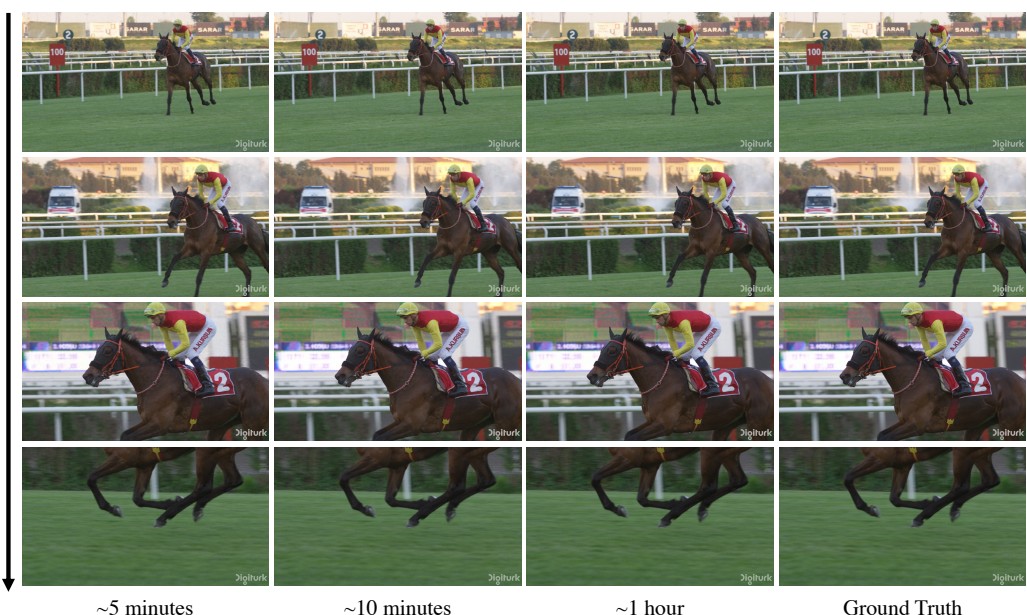

|  ~5 minutes | ~10 minutes | ~1 hour | Ground Truth |

Figure 15: Reconstruction results on Jockey in UVG-HD after training NVP-S for "5 minutes", "10 minutes", and "1 hour" with a single NVIDIA V100 32GB GPU.

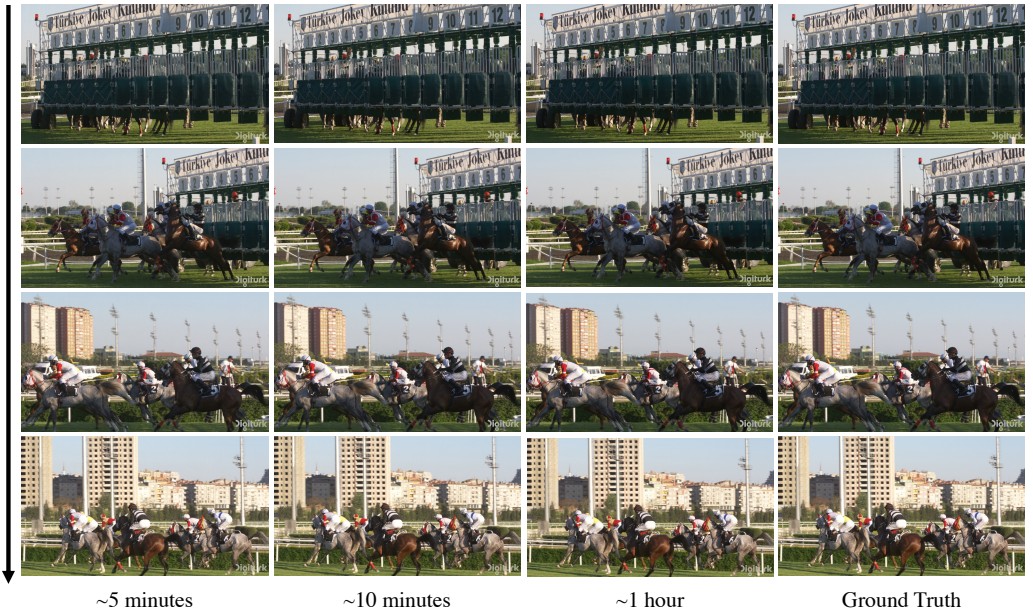

| ~5 minutes | ~10 minutes | ~1 hour | Ground Truth |

Figure 16: Reconstruction results on ReadySetGo in UVG-HD after training NVP-S for "5 minutes", "10 minutes", and "1 hour" with a single NVIDIA V100 32GB GPU.

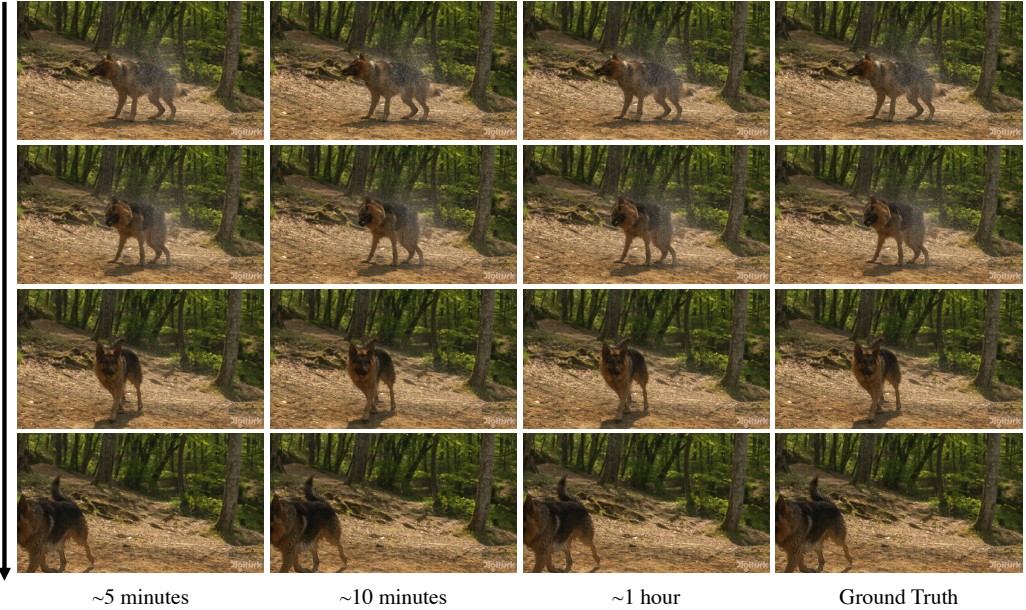

| ~5 minutes | ~10 minutes | ~1 hour | Ground Truth |

Figure 17: Reconstruction results on ShakeNDry in UVG-HD after training NVP-S for "5 minutes", "10 minutes", and "1 hour" with a single NVIDIA V100 32GB GPU.

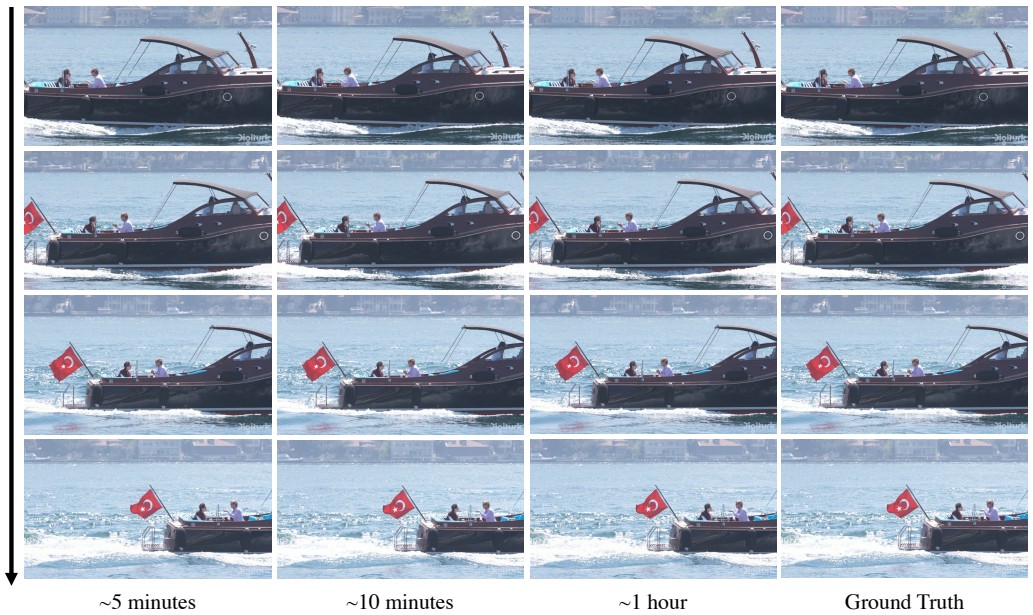

| ~5 minutes | ~10 minutes | ~1 hour | Ground Truth |

Figure 18: Reconstruction results on Yachtride in UVG-HD after training NVP-S for "5 minutes", "10 minutes", and "1 hour" with a single NVIDIA V100 32GB GPU.

## D  Comparison of video decoding time

We compare the decoding time of NVP with other baselines on a single GPU (NVIDIA V100 32GB).

Table 7: Decoding time of coordinate-based representations measured with FPS (higher is better). It was measured on Jockey (600 frames with $1920 \times 1080$ resolution) in UVG-HD with a single NVIDIA V100 32GB GPU.

| Model | BPP | FPS |
|---|---|---|
| Instant-ngp [34] | 7.352 | 29.81 |
| NeRV-S [5] | 0.177 | 45.39 |
| NeRV-L [5] | 0.426 | 15.28 |
| NVP-S | 0.172 | 6.51 |
| NVP-L | 0.359 | 4.87 |

NVP requires more decoding time than prior works, mainly due to (a) *three* learnable keyframes along each spatio-temporal axis (three times more access than the single grids of Instant-ngp [34]) and (b) the modulation architecture while evaluating the corresponding RGB values. However, note that the computation through each keyframe does not depend on each other; in this respect, the decoding time of NVP can be sped up significantly with a parallel design and implementation, *e.g.*, following the implementation details from Instant-ngp.[7] Moreover, while (b) exhibits considerable improvements in encoding complicated videos, we also demonstrate that our method still shows reasonable video encoding without modulations (both are shown in Section 4.3 of the main text); hence, one can control the trade-off between the decoding time and encoding quality by designating the multilayer perceptron (MLP) size and the whether the modulation is applied (or not).

---

[7]Unlike our method that utilizes the well-known Pytorch [36] framework, Instant-ngp utilizes C++/CUDA to implement *all* of the components for enabling strong parallelism in training and inference. As the official implementation of Instant-ngp[8] states that the decoding time highly deviates if one uses non-C++/CUDA frameworks, *e.g.*, Pytorch, we argue the gap of decoding time in Table 7 mainly stems from such different implementation details, and it can be remarkably mitigated by following their implementation.

[8]https://github.com/NVlabs/instant-ngp

# E More description and visualization of latent keyframes

Our keyframes aim to learn meaningful contents that are shared across a given video along an axis. For instance, the temporal axis ($\mathbf{U}_{\theta_{\mathrm{xy}}}$) learns common contents in every timeframe, such as background and the watermark in the video, which is invariant to timesteps.

The meanings of the other two keyframes ($\mathbf{U}_{\theta_{\mathrm{xt}}}$ and $\mathbf{U}_{\theta_{\mathrm{xt}}}$) may not be straightforward from their visualizations, as the shared contents across the other two directions in raw RGB space is often ambiguous. However, we note that we are learning the common contents in "latent space"; although they do not seem straightforward, they indeed play a crucial role in promoting the succinct parametrization of a given video (see Figure 8).

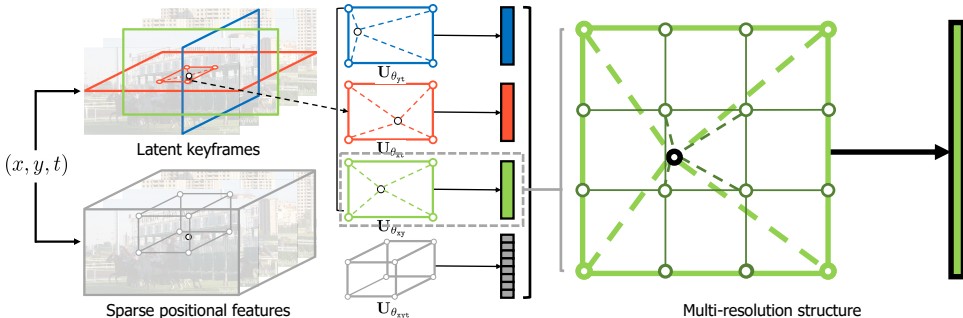

Figure 19: Illustration of our latent keyframe structure.

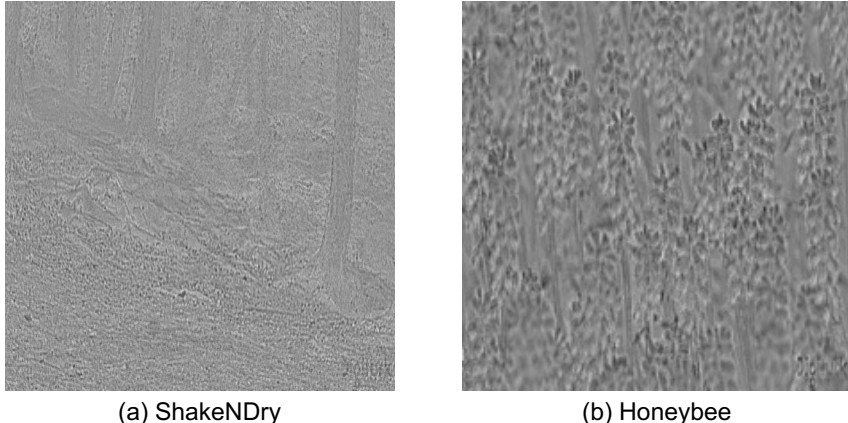

| (a) ShakeNDry | (b) Honeybee |

Figure 20: Illustration of our learnable latent keyframes $\mathbf{U}_{\theta_{\mathrm{xy}}}$ after encoding (a) ShakeNDry, and (b) Honeybee in UVG-HD, respectively.

# F Additional experiments on different datasets

## F.1 Experiment on Big Buck Bunny

Following the experimental setup in NeRV [5], we provide additional experimental results in the Big Buck Bunny video for a more intuitive comparison to prior work.

## F.2 Experiment with new videos

We also provide additional experimental results on more complex videos. In particular, we consider the following three new videos, where all of these videos are collected under the CC0 license:

- (Street) A timelapse video of a London street. People, cars, and buses are moving at different speeds as the traffic signal changes.[9]

---

[9] https://pixabay.com/videos/id-28693/

Table 8: PSNR values and encoding time of different CNRs to encode the Big Buck Bunny video. ↑ and ↓ denote higher and lower values are better, respectively.

| Method | BPP | PSNR (↑) | Encoding time (hr, ↓) |
|---|---|---|---|
| NeRV-S [5] | 0.128 | 32.36 | 1.734 |
| NVP-S (ours) | 0.136 | 32.56 | 0.925 |
| NeRV-M [5] | 0.249 | 36.50 | 1.762 |
| NVP-M (ours) | 0.248 | 36.49 | 0.925 |
| NeRV-L [5] | 0.496 | 39.26 | 1.774 |
| NVP-L (ours) | 0.456 | 39.88 | 0.925 |

- (City) A timelapse video of the city at night. A lot of cars are moving speedily.[10]
- (Surfing) A video of a man surfing in the ocean. Huge ocean waves are changing dramatically and fast.[11]

Table 9: PSNR values to encode more temporally complex videos.

| | UVG-HD (avg.) | Street | City | Surfing |
|---|---|---|---|---|
| BPP | 0.412 | 0.214 | 0.173 | 0.311 |
| PSNR | 37.71 | 38.90 | 38.45 | 43.71 |

---

[10]https://pixabay.com/videos/id-19627/
[11]https://pixabay.com/videos/id-110734/

# G   Video-wise compression result

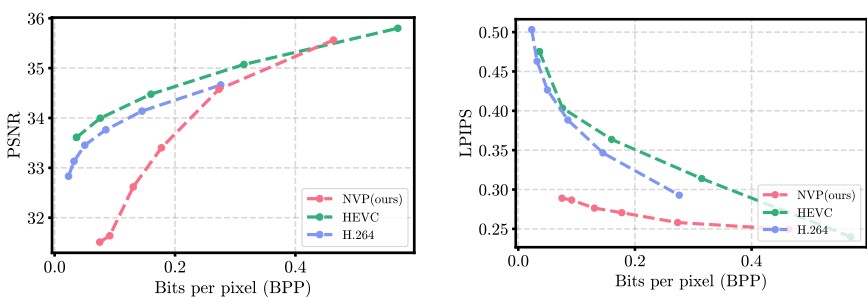

Figure 21: PSNR and LPIPS values of NVP and well-known video codecs over different BPP values computed on the Beauty video in UVG-HD.

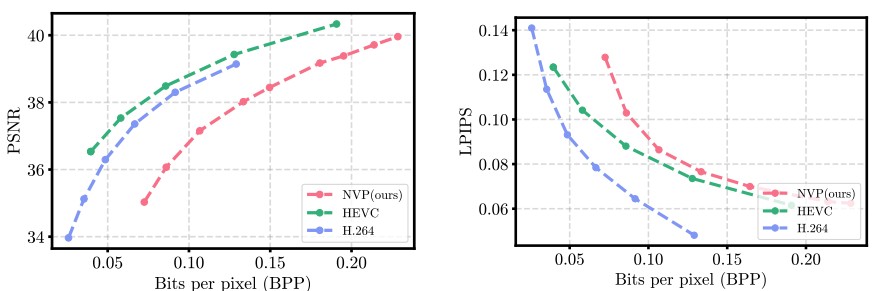

Figure 22: PSNR and LPIPS values of NVP and well-known video codecs over different BPP values computed on the Bosphorus video in UVG-HD.

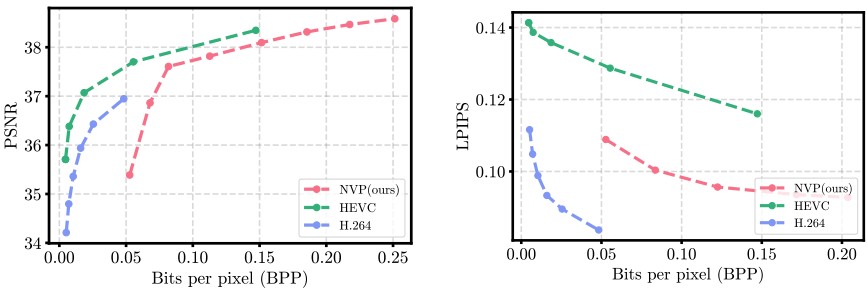

Figure 23: PSNR and LPIPS values of NVP and well-known video codecs over different BPP values computed on the Honeybee video in UVG-HD.

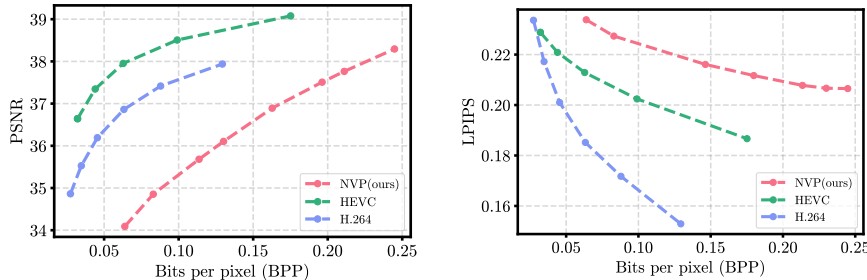

Figure 24: PSNR and LPIPS values of NVP and well-known video codecs over different BPP values computed on the Jockey video in UVG-HD.

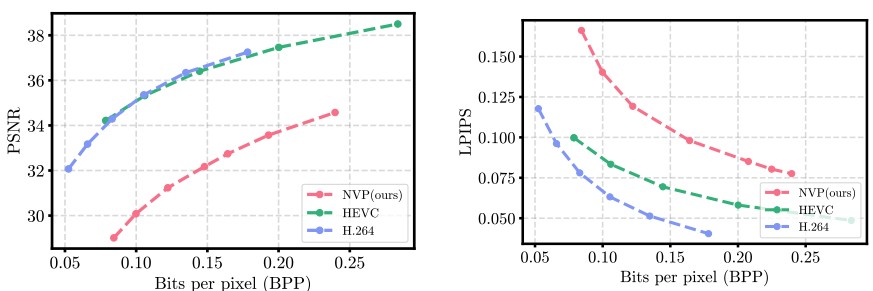

Figure 25: PSNR and LPIPS values of NVP and well-known video codecs over different BPP values computed on the ReadySetGo video in UVG-HD.

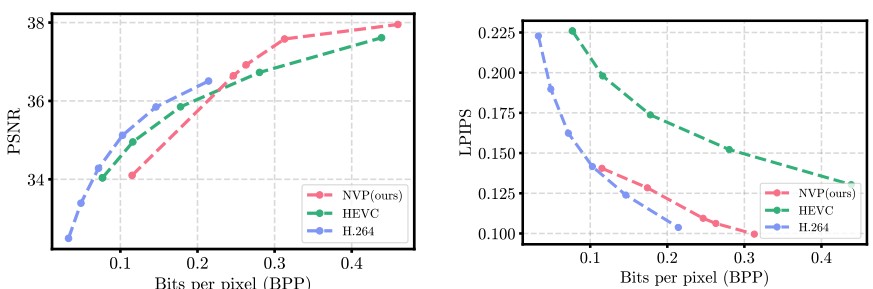

Figure 26: PSNR and LPIPS values of NVP and well-known video codecs over different BPP values computed on the ShakeNDry video in UVG-HD.

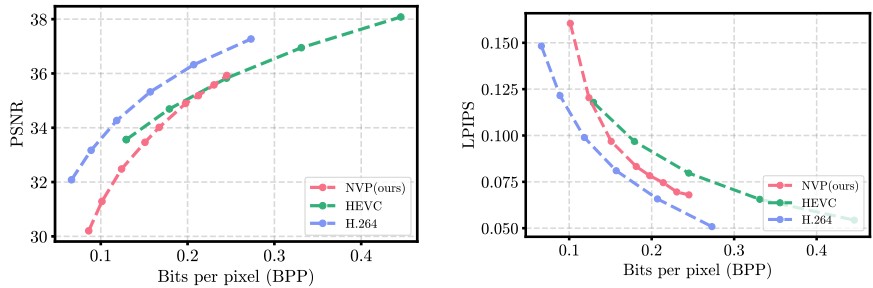

Figure 27: PSNR and LPIPS values of NVP and well-known video codecs over different BPP values computed on the Yachtride video in UVG-HD.

# H Comparison to learning-based video compression

In Figure 28, we compare compressed videos from NVP with well-known video codecs (H.264 [51], HEVC [43]) and state-of-the-art learning-based video compression methods (DVC [28], STAT-SSF-SP [56], HLVC [55], Scale-space [1], Liu et al. [25], and NeRV [5]) on UVG-HD. We train NVP for every single video separately, where we use the same architecture and hyperparameters for encoding and compressing all videos in UVG-HD.

Since our primary focus is not on video compression, there exists a gap between the existing compression methods and NVP. However, we strongly believe that there are numerous future directions to engage NVP for video compression, such as investigating per-video hyperparameter search strategy or designing better CNR architecture more specialized for compression.

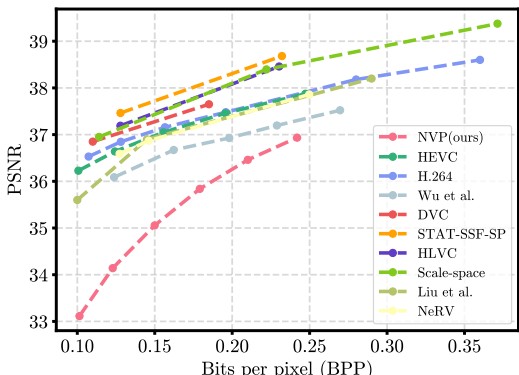

Figure 28: PSNR values of NVP, well-known video codecs, and learning-based video compression methods over different BPP values computed on UVG-HD.