# OpenReview forum: "Scalable Neural Video Representations with Learnable Positional Features"
_NeurIPS.cc/2022/Conference — NeurIPS 2022 Accept_

### Official Review · Reviewer_MBhC · 2022-07-11

**Rating:** 5
**Confidence:** 4
**Soundness:** 2 fair
**Presentation:** 3 good
**Contribution:** 2 fair

**Summary:**

This paper proposes a new CNR framework by introducing “learnable positional features” and decomposing the video into 2D and 3D latent representations. Besides, a compute- and memory-efficient compression procedure is introduced to further reduce the parameters and training cost. Experiments are conducted on UVG-HD benchmark for evaluating video encodings.

**Questions:**

Here are some concerns to be addressed:

1. My major concern about this work is the experiment results shown in Table. 1, in which the experimental setting is different from NeRV. Maybe the results on “Big Buck Bunny” sequence should be given to present a more intuitive comparison.
2. Also in Table. 1, why is instant-NGP not compared after encoding time > 15hours?
3. The results of video compression evaluation should be more detailed. Figure. 4 only shows the results on ShakeNDry and Beauty. More comprehensive experimental results should be presented.
4. Lack of comparison of decoding time.
5. Why is the grid size selected as H×W×S? It is better to show the influence of different grid sizes.
6. More experiments comparing with traditional coding frameworks such as VVC and HEVC are suggested.


**Limitations:**

Yes.

**Strengths And Weaknesses:**

Strengths:
This work proposes a new CNR framework to solve the dilemma of quality and efficiency in video encoding. The experimental results have been improved on both PSNR metrics and parameter-efficiency. The overall clarity of the paper is good.

Weakness:
This work is mainly built based on the existing CNR framework, but the overall originality is not high. The experimental part is not sufficient, and some experimental settings are also unfair to some extent.

---

> ### Author Response · Authors · 2022-08-02
> **Response to Reviewer MBhC (3/3)**
>
> **[Q4] Lack of comparison of decoding time.**
>
> In our original submission, we indeed included the comparison of decoding time in Appendix D (we revised the manuscript to point out the comparison in L244): for instance, measured with a single NVIDIA V100 32GB GPU, NeRV-L decodes 15.28 frames per second (FPS), while our current Pytorch implementation of NVP-L (our method) decodes 5.71 FPS. Yet, we strongly believe that such a gap can be overcome via a C++/CUDA-based implementation of our method. Specifically, Instant-ngp [2] exhibits the decoding speed of CNRs can be remarkably boosted (40.10 FPS measured with the above setup) via C++/CUDA-based parallelism implementation if the architecture follows a combination of latent grids and a simple multi-layer perceptron (MLP). Recall that NVP is also composed of latent grids and MLP; we strongly believe the decoding speed of NVP can be comparable to NeRV [1] if one follows the implementation details in Instant-ngp. We also note that such parallelism is not straightforward for NeRV, which uses a convolutional neural network for its architecture. Furthermore, in contrast to NeRV, NVP synthesizes _each pixel value independently_ which does not require decoding of the entire frame at once, providing better scalability on training and inference at extremely high-resolution videos (e.g., 8K videos) under limited memory constraints. We will provide the C++/CUDA-based implementation of NVP in the final version of our manuscript.
>
> ---
>
> **[Q5] Explanation about the choice of grid size.**
>
> To answer your question, we provide the result of the ReadySetGo video measured with the PSNR metric (higher is better) under the different configurations $(H, W, S)$ of the sparse positional features. Here, we set $H \times W \times S$ similarly for a fair comparison among different hyperparameter setups. As shown in the below table, our method is quite robust to the selection of $(H,W,S)$. Moreover, one can observe that letting large $S$ is more beneficial: this is because (a) we select $3 \times 3 \times 1$ latent codes from sparse positional features and (b) the spatial keyframe often becomes more informative by capturing the common content across the temporal axis (e.g., background) and can dramatically reduce the value of $H$ and $W$.
>
> \begin{array}{l c}
> \hline
> (H, W, S) & \text{PSNR }(\uparrow) \newline
> \hline
> (300, 300, 600) & 36.72 & \newline
> (450, 450, 300) & 35.87 & \newline
> (225, 400, 600) & 36.64 & \newline
> \hline
> \end{array}
>
> ---
>
> [1] Chen et al., NeRV: Neural Representations for Videos, NeurIPS 2021
> [2] Müller et al., Instant Neural Graphics Primitives with a Multiresolution Hash Encoding, SIGGRAPH 2022
> [3] UVG Dataset: 50/120fps 4K Sequences for Video Codec Analysis and Development, MMSys 2020
> [4] Lu et al., DVC: An End-to-End Deep Video Compression Framework, CVPR 2019
> [5] Agustsson et al., Scale-space flow for end-to-end optimized video compression, CVPR 2020

---

> > ### Author Response · Authors · 2022-08-08
> > **A gentle reminder for Reviewer MBhC**
> >
> > Dear Reviewer MBhC,
> >
> > We again sincerely appreciate your efforts and time in reviewing and providing incisive comments on our paper.
> >
> > We kindly remind you that the discussion period will end in two days. We believe that we successfully addressed your concerns, questions, and suggestions with the results of the supporting experiments and the revised manuscript.
> >
> > If you have any further concerns, questions or suggestions, please do not hesitate to let us know.
> >
> > Thank you very much!
> >
> > Authors

---

> > ### Comment · Reviewer_MBhC · 2022-08-09
> > **Response to rebuttal**
> >
> > Many thanks to authors who provide more detailed experimental results and have refined overall writting following reviewers' comments. The response from authors well clarifies most of my concerns and I want to keep my initial decision of acceptance. If possible I want to see more competing results with deep video codec under a fair evaluation condition, although I understand that compression is not the major goal.

---

> > > ### Author Response · Authors · 2022-08-09
> > > **Response to Reviewer MBhC**
> > >
> > > Thank you for your response! We are happy to find that our rebuttal successfully addressed all of your concerns. Although our primary focus is not video compression, we also think that providing the comparison results with deep video codecs under the fair evaluation condition would further strengthen our paper and give a good insight into future research in both CNRs and video compression. Thank you for the suggestion, and we will add such a comparison in the final version of the manuscript.

---

> ### Author Response · Authors · 2022-08-02
> **Response to Reviewer MBhC (2/3)**
>
> **[Q3/6] More comprehensive compression results (video-wise result, traditional codecs as baselines).**
>
> We remind you that our primary goal is to design a better coordinate-based neural representation (CNR) for videos that is highly parameter-/compute-efficient and provides high-quality encoding, not to develop a video compression method. Video compression is just one of the possible use cases of CNRs, such as video inpainting, video frame interpolation, and super-resolution. Hence, we did not perform an extensive evaluation of the compression performance that considers lots of state-of-the-art video codecs as baselines. However, we want to note that we already have included HEVC (that you mentioned) and H.264, which are current advanced video codecs, to validate the potential of our approach for compression (see Figure 6 in the manuscript). Moreover, we also agree the more detailed video-wise compression results would further strengthen our manuscript; following your suggestion, we added the video-wise compression results on all the videos in the UVG benchmark in Appendix G.
>
> Although our main focus is not on video compression, we think investigating our work further for compression is worth it, as our approach has the potential to mitigate the limitations of learning-based compression schemes. Recall that most learning-based video compression methods learn the compression procedure from video “datasets” (e.g., [4]), while our method requires and uses only a “single test video” for representing a video as succinct parameters. Hence, while existing methods mostly suffer from the distributional shift [5], our approach does not, since it works in a “zero-shot” manner. Not limited to, compared with traditional codecs, our method can provide other intriguing properties that the existing codecs cannot achieve, e.g., video inpainting, video frame interpolation, super-resolution, etc. We think finding a better method of video compression based on our work should be definitely an interesting future direction.

---

> ### Author Response · Authors · 2022-08-02
> **Response to Reviewer MBhC (1/3)**
>
> We deeply appreciate your thoughtful comments and efforts in reviewing our manuscript. We mark our major revision in the revised manuscript with “blue.” We also illustrate several additional images and videos on our project page to answer your questions: https://neurips2022-nvp.github.io. We respond to each of your comments one-by-one in what follows.
>
> ---
>
> **[W1] Overall originality seems not high.**
>
> As highlighted by Reviewer kkNU, we believe that we propose various novel ideas, e.g., learnable latent keyframes across each spatio-temporal axis and the compression procedure. More importantly, we think that the originality of our method can be found not only in its detailed technical components but also in its remarkable goal. Namely, differentiated from previous CNRs, our method, NVP, is the first video CNR that achieves all three desired aspects of parameter-, compute-efficiency, and high-quality encoding simultaneously in encoding videos. Specifically, prior works have sacrificed one of them: for instance, NeRV [1] requires a very long encoding time to encode a given video, and Instant-ngp [2] requires lots of memory than the original video size to accomplish the compute-efficiency. In this respect, we think that we are tackling and solving an important problem of high originality.
>
> ---
>
> **[W2] Unfair, insufficient experimental setups in some aspects.**
>
> For a fair comparison, we use the official implementations of all of the baselines, following their exact training specification and hyperparameter setups, as stated in Appendix A. We also provide a detailed answer and more experimental results to address your concern on experimental setups; please refer to [Q1] and [Q2] below.
>
> ---
>
> **[Q1] Different experimental setup (Table 1) from NeRV.**
>
> We mainly conduct the experiments under the UVG benchmark since we believe UVG can provide a more extensive comparison than a single Big Buck Bunny video used in NeRV [1]. Specifically, UVG contains “multiple” videos, where each video has different characteristics (e.g,, motion and texture) [3] consisting of frames of higher resolution and a longer length than the Big Buck Bunny video. For example, the resolution of videos in UVG datasets are 1920$\times$1080 (2.6$\times$ larger than Big Buck Bunny) and mostly have a length of 600 ($\times$4.5 times longer than Big Buck Bunny). Moreover, in contrast to NeRV, which only reports the encoding quality at the convergence, we report the result at various encoding times; this is because compute-efficiency is one of our major criteria to evaluate CNRs, not only their final performance and parameter-efficiency.
>
> Nevertheless, we also agree that providing a result on Big Buck Bunny can present a more intuitive comparison to NeRV and make the experiment more comprehensive. Thus, we present the comparison result of NeRV and our method on the Big Buck Bunny video (we added this in Appendix F) in the table below, which validates the superiority (or at least comparable) of our method in the setup of NeRV.
>
> \begin{array}{l ccc}
> \hline
> \text{Method} & \text{BPP} & \text{PSNR } (\uparrow)  & \text{Encoding time } (\text{hr}, \downarrow) \newline
> \hline
> \text{NeRV-S} & 0.128 & 32.36 & 1.734\newline
> \text{NVP-S (ours)} & 0.136 & 32.56 & 0.925 \newline
> \hline
> \text{NeRV-M} & 0.249 & 36.50 & 1.762 \newline
> \text{NVP-M (ours)} & 0.248 & 36.49 & 0.925\newline
> \hline
> \text{NeRV-L} & 0.496 & 39.26 & 1.774 \newline
> \text{NVP-L (ours)} & 0.456 & 39.88 & 0.925\newline
> \hline
> \end{array}
>
> ---
>
> **[Q2] Missing the results of instant-ngp in Table 1 after training >15 hours.**
>
> Our main focus of >15 hours encoding time in Table 1 is to compare the parameter efficiency of different methods, while Instant-ngp [2] is proposed to sacrifice its parameter-efficiency for achieving high-quality encodings. Thus, we do not consider Instant-ngp in this situation. Nonetheless, to further alleviate your concern, we provide the results of Instant-ngp after training >15 hours, where the model size is adjusted similarly to other baselines. As shown in the below table, our method shows better results even compared with Instant-ngp in these setups. We also added this result in Table 1 and clarified the intention of each row.
>
> \begin{array}{clccc}
> \hline
> \text{Encoding time}&\text{Method}&\text{BPP}&\text{PSNR } (\uparrow)&\text{FLIP }(\downarrow) &  \text{LPIPS }(\downarrow) \newline
> \hline
> \text{>15 hours} & \text{Instant-ngp} & 0.229 & {28.81\small{\pm3.48}} & {0.155\small{\pm0.057}} & {0.390\small{\pm{0.135}}} \newline
> \sim\text{12 hours}& \text{NVP-S (ours)}&0.214 & \mathbf{36.34\small{\pm2.19}} & \mathbf{0.067\small{\pm0.017}} & \mathbf{0.128\small{\pm{0.073}}}\newline
> \hline
> \text{>40 hours}&\text{Instant-ngp}&0.436&29.98\small{\pm3.39}&0.138\small{\pm0.051}&{0.358\small{\pm{0.140}}}\newline
> \sim\text{17 hours}&\text{NVP-L (ours)}&0.412&\mathbf{37.71\small{\pm1.80}}&\mathbf{0.061\small{\pm0.013}}& \mathbf{0.110\small{\pm{0.085}}}\newline
> \hline
> \end{array}

---

### Official Review · Reviewer_kkNU · 2022-07-12

**Rating:** 6
**Confidence:** 3
**Soundness:** 3 good
**Presentation:** 2 fair
**Contribution:** 3 good

**Summary:**

The paper proposes a coordinate-based architecture for video representation. The key element of the architecture is the introduction of keyframes (positional encodings of the features). The approach is validated on the UVG benchmark showing benefits over prior art.

**Questions:**

How does the method scale with video size in terms of both the spatial resolution of the frame as well as the number of frames?

**Limitations:**

The limitations are discussed but are centered around the general idea of CNR for video representation. The discussion of the possible limitation of the model design could be extended. For example, one might expect that the keyframe idea (amortizing over the video) would not work well for long videos composed of multiple possibly dynamic scenes.



**Strengths And Weaknesses:**

The paper is in general well-structured and well written. The model design choices are motivated, and the reported results looks solid. The idea is moderately original and the quality of the work is good enough to recommend the acceptance. However,  he clarity of the presentation could be improved and the authors could highlight better the significance of the work. See detailed comments and questions below.

**Abstract:**
- Abstract could be strengthened to reflect better the paper contribution and its importance.

**Introduction:**
- The introductory section positions well the work w.r.t prior art.
- Introduction does not motivate why research on CNRs for video representation is an important research avenue. Adding such motivation would make the paper stronger.
- The connection between learnable positional features and keyframes is now well exposed in the intro and in the abstract. It is hard to understand why keyframes could play the role of learnable positional features.
- Fig 2 could be a bit more informative; the key element of the paper seems to be the construction of the keyframes however, the figure does not give details on how the keyframes are constructed.
- L24 “… numerous appealing properties …“- only one property is mentioned.
- L36 “… while enjoying lots of intriguing properties …” – could the authors enumerate the properties
- L71 “… 34.43 in 5 minutes …” – can the authors add details here? What is the hardware? How is the time measured?



**Methodology:**
- In general methodology section is well written and easy to follow. One exception is the construction of the keyframes. This part could benefit from some re-writing. Also, adding a figure explaining the process could ease the understanding. Moreover, it is unclear why U_l is represented as a set.
- Did the authors consider different types of or designs of keyframes?
- Modulation, in L174 the authors write: “… we found such a simple MLP architecture lacks an expressive power and fails to capture the complex dynamics… “. However, based on ablation study the benefit of modulation seems marginal (in terms of PSNR). Could the authors comment on this?

**Experiments:**
- The ablation table is missing std, could the std information be added to the ablation table?
- Adding additional dataset would make the validation and observations stronger.

---

> ### Author Response · Authors · 2022-08-02
> **Response to Reviewer kkNU (2/2)**
>
> **[W7] More experiments on additional datasets.**
>
> Thank you for your constructive comment. Following your suggestion, we provide additional experimental results on a Big Buck Bunny video (following setups in NeRV [1]) and more complex videos. In particular, we consider the following three new videos:
> - (Street) A timelapse video of a London street. People, cars, and buses are moving at different speeds as the traffic signal changes [3].
> - (City) A timelapse video of the city at night. Lots of cars are moving speedily [4].
> - (Surfing) A video of a man surfing in the ocean. Huge ocean waves are changing dramatically and fast [5].
>
> As shown in the below table, our method also shows similar, or even better results in different, diverse datasets. We also provide the visualization of these videos on our project page and added Appendix F in the revision to include these additional experiments.
>
> \begin{array}{c ccccc}
> \hline
> & \text{UVG-HD (avg.)} & \text{Street} & \text{City} & \text{Surfing} & \text{Big Buck Bunny}  \newline
> \hline
> \text{BPP} & 0.412 & 0.214 & 0.173 & 0.311 & 0.456  \newline
> \text{PSNR} & 37.71 & 38.90 & 38.45 & 43.71 & 39.88 \newline
> \hline
> \end{array}
>
> ---
>
> **[W8] Further discussion about possible limitations of our method.**
>
> Thank you for your incisive comment. As you stated, applying our model to an extremely long video may not work well, as they include multiple dynamic scenes which are not highly related to each other and thus make it challenging to learn the common contents. Developing a better form of latent keyframes suitable for much longer videos should be one of the important future directions of our work. One of the naive (but strong) baselines can be considering multiple keyframes per each axis to learn shared contents at each "part" of a given video, rather than having a single keyframe to capture the common content in the whole video.
>
> Moreover, as our primary focus is not video compression, the current compression performance is sometimes below the state-of-the-art video codecs (e.g., HEVC) or learning-based compression method (e.g., [6]). However, we think investigating our work further for compression is worth it, as our approach has the potential to mitigate the limitations of learning-based compression schemes. Recall that most learning-based video compression methods learn the compression procedure from video “datasets” (e.g., [6]), while our method requires and uses only a “single test video” for representing a video as succinct parameters. Hence, while existing methods mostly suffer from the distributional shift [7], our approach does not, since it works in a “zero-shot” manner. Not limited to, compared with traditional codecs, our method can provide other intriguing properties that the existing codecs cannot achieve, e.g., video inpainting, video frame interpolation, super-resolution, etc. We think finding a better method of video compression based on our work should be definitely an interesting future direction.
>
> ---
>
> **[Q1] Scalability of NVP with the large-scale videos (spatial resolution & the number of frames).**
>
> Our method, NVP, can be scaled up to more large-scale videos that consist of higher resolution and more frames. To validate the scalability of NVP, we provide an additional experiment on a video consisting of 1200 frames of 3840$\times$2560 resolution, 4$\times$ higher resolution, and 2$\times$ to 4$\times$ longer than the videos in the UVG benchmark. Measured with the PSNR metric (higher is better), NVP exceeds 37.77 with 12 hours (encoding time) and 0.130 (BPP), which is even better than the average result on the UVG benchmark (36.34 with 12 hours (encoding time) and 0.214 (BPP)). We also provide the qualitative results of this video on our project page.
>
> ---
>
> [1] Chen et al., NeRV: Neural Representations for Videos, NeurIPS 2021
> [2] Müller et al., Instant Neural Graphics Primitives with a Mulriesolution Hash Encoding, SIGGRAPH 2022
> [3] https://pixabay.com/videos/id-28693/
> [4] https://pixabay.com/videos/id-19627/
> [5] https://pixabay.com/videos/id-110734/
> [6] Lu et al., DVC: An End-to-End Deep Video Compression Framework, CVPR 2019
> [7] Agustsson et al., Scale-space flow for end-to-end optimized video compression, CVPR 2020

---

> > ### Author Response · Authors · 2022-08-08
> > **A gentle reminder for Reviewer kkNU**
> >
> > Dear Reviewer kkNU,
> >
> > We again sincerely appreciate your efforts and time in reviewing and providing incisive comments on our paper.
> >
> > We kindly remind you that the discussion period will end in two days. We believe that we successfully addressed your concerns, questions, and suggestions with the results of the supporting experiments and the revised manuscript.
> >
> > If you have any further concerns, questions or suggestions, please do not hesitate to let us know.
> >
> > Thank you very much!
> >
> > Authors

---

> ### Author Response · Authors · 2022-08-02
> **Response to Reviewer kkNU (1/2)**
>
> We deeply appreciate your thoughtful comments and efforts in reviewing our manuscript. We mark our major revision in the revised manuscript with “blue.” We also illustrate several additional images and videos on our project page to answer your questions: https://neurips2022-nvp.github.io. We respond to each of your comments one-by-one in what follows.
>
> ---
>
> **[W1] Editorial comments on Abstract, Introduction, and Method.**
>
> Thank you for your insightful comment! As you suggested, we improved the writing and illustration as follows:
> - (Abstract) We explicitly highlight the role of latent keyframes and the contribution of our method.
> - (Introduction) We enumerate the numerous intriguing properties of CNRs in general (L26-27) and video CNRs (L32) so that readers better understand the motivation for developing video CNRs.
> - (Method) We re-write several sentences (e.g., L134-135) on latent keyframes and fix the set notation to prevent confusion.
>
> ---
>
> **[W2] How, where is the “5 min” in Introduction measured?**
>
> Thank you for pointing this out. It is evaluated under a single V100 32GB GPU and 28 instances from a virtual CPU of Intel Xeon Platinum 8168 GPU, where the time is measured from the time when the first iteration starts for each of the baselines. We also clarified this in the revision (L74).
>
> ---
>
> **[W3] Figure 2 can be more informative to give details on the construction of the keyframes.**
>
> Thank you for the suggestion. We add Figure 17 to illustrate how the latent keyframes are constructed, and mention this figure in the caption of Figure 2.
>
> ---
>
> **[W4] Did the authors try different design choices of keyframes?**
>
> We tried several different operations to compute the latent features from a given coordinate, and the latent grids (e.g., bicubic, nearest operations), but the current choice worked best. Still, there are many potential directions of developing keyframes one can explore, e.g., considering latent keyframes across axes other than spatio-temporal axes may increase the expressive power of the overall framework and would become an interesting direction.
>
> ---
>
> **[W5] Improvement from modulation seems marginal .**
>
>
>
>
> We agree the ablation study on a single Jockey video (in the initial manuscript) may mislead you to think the improvement is marginal. However, modulation indeed provides considerable improvement; on average of all videos in the UVG benchmark, modulation improves the PSNR metric (higher is better) from 38.04$\rightarrow$38.89 (we update this in the revision). In addition, when used alone, modulation dramatically improves the compute-efficiency: to validate this, we provide an additional ablation study on modulation without sparse positional features on Beauty video in UVG-HD. As shown in the below table, our method reaches 30.68 (measured with PSNR metric; higher is better) with the modulation, which is >15.01$\times$ compute-efficient than the model without modulation.
>
> \begin{array}{l c c c}
> \hline
> \text{Method} & \text{\\# Params.} & \text{PSNR } (\uparrow) & \text{Training time} \newline
> \hline
> \text{Latent keyframes} & \text{138M} & 30.66 &1276\text{s} \newline
> \text{Latent keyframes + Modulation} & \text{138M} & 30.68 & 85\text{s}\newline
> \hline
> \end{array}
>
>
> ---
>
> **[W6] Adding standard deviation in ablation study (Table 3).**
>
> In the initial manuscript, we did not report standard deviations as we conducted the ablation study with a single video in the UVG benchmark (Jockey), a video in which the temporal variation is large. As you highlighted, we performed the ablation study with all 7 videos and added the standard deviation with the average and updated Table 3 like the below table in the revision.
>
> \begin{array}{c c c c c}
> \hline
> \text{Keyframes} & \text{Sparse feat.} & \text{Module.}  & \text{\\# Params.} & \text{PSNR }(\uparrow) \newline
> \hline
> \color{red}\tt{X} & \color{green}\checkmark & \color{green}\checkmark & \text{136M} & 31.21\small{\pm2.80} \newline
> \color{green}\checkmark & \color{red}\tt{X} &  \color{green}\checkmark & \text{138M} & 32.44\small{\pm4.48}\newline
> \color{green}\checkmark &  \color{green}\checkmark & \color{red}\tt{X} & \text{147M} & 38.04\small{\pm2.27}\newline
> \hline
> \color{green}\checkmark &  \color{green}\checkmark & \color{green}\checkmark & \text{136M} & \mathbf{38.89\small{\pm2.11}}\newline
> \hline
> \end{array}

---

### Official Review · Reviewer_LDYR · 2022-07-14

**Rating:** 6
**Confidence:** 4
**Soundness:** 2 fair
**Presentation:** 2 fair
**Contribution:** 2 fair

**Summary:**

The authors proposed a scalable implicit neural network for reconstructing video signals in a parameter-efficient and computationally efficient manner. In order to do this, they proposed learnable positional features inspired by the existing work in the literature (Instant-ngp). In general, implicit neural network maps from input co-ordinates to RGB pixels directly using one mapping function, here the authors maps the pixel co-ordinates to intermediate latent space, and from latent space to RGB pixels using another mapping function with modulation function. In the latent space, the authors learn the key frames of the given video in the latent grids, for xy, xt, yt, xyt. These are compressed using existing standard codecs HEVC and JPEG, and interpolation is used to find the latent vectors for any given (x,y,t), and then latent to RGB mapping function. The authors show their proposed method trains faster, and reconstructs the video better in a parameter efficient way.

**Questions:**

1) In learning latent keyframes, what kind of information do the keyframes contains, and in terms of visualization how does it look like?. does it learn image like structure?
2) how the BPP is calculated? whether the weights of the network are used in full precision or half precision to compute the BPP. In the total filesize, how much information is from latent grids, and how much information is from the latent to RGB mapping function.
3) I visualized a few videos in the dataset used in the paper, the temporal motion in the videos is not much dynamic, it's almost static. Maybe this is the reason, the method is able to have good reconstruction results. I suspect whether the proposed might have a similar performance with high temporal variations.


**Limitations:**

the limitations are discussed in the paper

**Strengths And Weaknesses:**

Strengths:
The paper proposes the parameter and computational efficient implicit neural network to reconstruct videos from pixel coordinates. The method has the advantage of learning keyframes automatically from the videos, whereas the other compression methods do not have the capability to learn the keyframes.
Learns the grid representations in the latent space for different directions (xy, xt, yt, xyt). The authors claim that it improves parameter efficiency of Neural video representation (NVR)

Weakness:
For the compression of the latent spatial grids, the HEVC and JPEG compression methods are used. For the video compression applications, the combination of traditional codecs with learning-based compression is not interesting in my opinion. The traditional codecs are bound to have artifacts that the proposed method incorporates into the proposed method
The performance of the proposed method are below the HEVC and H264, and the datasets used to evaluate the proposed method is not a good benchmark. I see that in the honeybee video, the video is almost static and it is difficult to conclude whether the proposed method can capture the dynamic motions in the temporal dimensions.
The proposed method is not compared with other state-of-the-art methods in learning-based video compression methods.

---

> ### Author Response · Authors · 2022-08-02
> **Response to Reviewer LDYR (2/2)**
>
>
> **[Q1] Visualization and the information included in learnable latent keyframes.**
>
> Our keyframes aim to learn meaningful contents that are shared across a given video along an axis (we added visualization of these keyframes in Appendix E). For instance, the temporal axis ($\mathbf{U}_{\theta_\mathtt{xy}}$) learns common contents in every timeframe, such as background and the watermark in the video which is invariant to timesteps.
>
> The meanings of the other two keyframes ($\mathbf{U\}_{\theta_\mathtt{xt}}$ and $\mathbf{U}\_{\theta_\mathtt{xt}}$) may not be straightforward from their visualizations, as the shared contents across the other two directions in raw RGB space is often ambiguous. However, we note that we are learning the common contents in “latent space”; although they do not seem straightforward, they indeed play a crucial role in promoting the succinct parametrization of a given video. We also performed the ablation study to show the effect of these two keyframes; please refer to Figure 8 for the result.
>
> ---
>
> **[Q2] How the BPP is calculated and the portions of bits of each component?**
>
> We use 32-bit precision for modulated implicit function and 8-bit precision for learnable latent keyframes and sparse positional features: we consider such different precisions in calculating the BPP value of our method. The 8-bit precision of learnable latent keyframes and sparse positional features originates from their compression procedure of them after training, as we quantize them as 8-bit precision and leverage traditional image and video codecs (respectively) to reduce their size. Here, we note that we train the model using 32-bit precision for all latent features and the modulated implicit function before the compression.
>
> After operating our proposed compression scheme, the number of bits (or information) is mostly in the order of sparse positional features, learnable keyframes, and the modulated implicit function. For instance, for encoding Yachtride video with NVP-L model, each component uses the following number of bits (reported with kilobyte (kB):
> - Learnable latent keyframes: 18,147kB (30.8%)
> - Sparse positional features: 40,070kB (68.1%)
> - Modulated implicit function: 617kB (0.01%)
>
> ---
>
> **[Q3] Does the proposed method achieve similar performance on videos with high temporal variations?**
>
> Our method achieves similar performance on videos with high temporal variations. As mentioned above (in [W3]), the UVG benchmark contains videos with dynamic motions, e.g., Jockey and ShakeNDry. On the project page (see also Figure 1), we exhibit how well our method, NVP, encodes Jockey with high quality while simultaneously achieving compute-/parameter-efficiency. We also exhibit on the webpage how well NVP succinctly encodes a ShakeNDry without suffering from any artifacts (see also Figure 3).
>
> Nonetheless, to further address your concern, we provide the video-wise quantitative result of Jockey and ShakeNDry videos as well as the result of other videos not in the UVG benchmark that contains dynamic, complex motions by conducting additional experiments. In particular, we consider the following three new videos:
> - (Street) A timelapse video of a London street. People, cars, and buses are moving at different speeds as the traffic signal changes [3].
> - (City) A timelapse video of the city at night. Lots of cars are moving speedily [4].
> - (Surfing) A video of a man surfing in the ocean. Huge ocean waves are changing dramatically and fast [5].
>
> As shown in the below table, our method also shows similar, or even better results in different, diverse datasets which are much more dynamic than the Honeybee video, validating the effectiveness of our method regardless of temporal variations. We also provide the visualization of these videos on our project page and added Appendix F in the revision to include these additional experiments.
>
> \begin{array}{l cccccc}
> \hline
> & \text{Honeybee (static)} & \text{Jockey} & \text{ShakeNDry} & \text{Street} & \text{City} & \text{Surfing} \newline
> \hline
> \text{BPP} & 0.409 & 0.325 & 0.305  & 0.214 & 0.173 & 0.311 \newline
> \text{PSNR} & 38.58 & 38.19 & 37.67 & 38.90 & 38.45 & 43.71 \newline
> \hline
> \end{array}
>
> ---
>
> [1] Lu et al., DVC: An End-to-End Deep Video Compression Framework, CVPR 2019
> [2] Agustsson et al., Scale-space flow for end-to-end optimized video compression, CVPR 2020
> [3] https://pixabay.com/videos/id-28693/
> [4] https://pixabay.com/videos/id-19627/
> [5] https://pixabay.com/videos/id-110734/

---

> > ### Author Response · Authors · 2022-08-08
> > **A gentle reminder for Reviewer LDYR**
> >
> > Dear Reviewer LDYR,
> >
> > We again sincerely appreciate your efforts and time in reviewing and providing incisive comments on our paper.
> >
> > We kindly remind you that the discussion period will end in two days. We believe that we successfully addressed your concerns, questions, and suggestions with the results of the supporting experiments and the revised manuscript.
> >
> > If you have any further concerns, questions or suggestions, please do not hesitate to let us know.
> >
> > Thank you very much!
> >
> > Authors

---

> ### Author Response · Authors · 2022-08-02
> **Response to Reviewer LDYR (1/2)**
>
> We deeply appreciate your thoughtful comments and efforts in reviewing our manuscript. We mark our major revision in the revised manuscript with “blue.” We also illustrate several additional images and videos on our project page to answer your questions: https://neurips2022-nvp.github.io. We respond to each of your comments one-by-one in what follows.
>
> ---
>
> **[W1] Combining traditional codecs is not compelling due to artifacts and bounds of the codecs on compression.**
>
> We first emphasize that the primary goal of ours is NOT to develop a better compression method. Instead, we aim for designing a better coordinate-based neural representation (CNR) for videos that is highly parameter-/compute-efficient and provides high-quality encoding. Here, video compression is just one of many possible use cases of CNRs, such as video inpainting, video frame interpolation, super-resolution, etc.
>
> There are indeed several works that exploited CNRs for the purpose of compression. What our paper asks, on the other hand, is somewhat the opposite: can we use traditional codecs to design a better CNR encoding scheme? In this respect, we rather think incorporating traditional codecs into video CNRs is a novel and strong feature of our method, not existing in other works. Moreover, combining these codecs and video CNRs even has the potential to remove the artifacts caused by traditional codecs, making this direction appealing even from the perspective of compression. For instance, Figure 9 shows that traditional codecs often suffer from the artifact of inconsistent encoding quality over different timesteps, whereas our method (combined with traditional codecs) achieves consistent encoding performance and mitigates such artifacts.
>
> ---
>
> **[W2] The compression performance of NVP is worse than the traditional codecs.**
>
> We agree that the compression performance of our method, NVP, is sometimes below state-of-the-art traditional codecs (e.g., H.264, HEVC) with respect to PSNR metrics. However, we remark that our method shows a better perceptual similarity (measured with the LPIPS metric) than traditional codecs (as shown in Figure 6). Hence, it is arguable to say which method is clearly superior to the other baselines. Moreover, as we previously mentioned (in [W1]), we would like to emphasize that our main focus is not video compression; our method provides other intriguing properties that the existing codecs cannot achieve, e.g., video inpainting, video frame interpolation, and super-resolution, etc. To clarify our main focus, we revised the manuscript (Section 1 and 3) to clarify our main goal and emphasize the benefits of CNRs. We also revised Section 4.2 to demonstrate various applications of our method as video CNRs. In particular, we additionally provide a video inpainting result (Figure 4) and a super-resolution result (Figure 5) of NVP; please refer to our project page for better visualization.
>
> ---
>
> **[W3] The benchmark dataset (UVG) seems to mostly contain static videos and thus not a good benchmark.**
>
> We first note that the UVG dataset also contains videos that are much more temporally dynamic than “Honeybee” (that you mentioned), such as “Jockey” or “ShakeNDry”: “Jockey” is a video of running horses, where the background and the poses of horses change rapidly (see the project webpage above for an excerpt); “ShakeNDry” is a video of an animal shaking its body and walking out of the camera frame. In this respect, we believe that the UVG dataset can be a more meaningful benchmark to evaluate video encoding.
>
>
> Nevertheless, following your suggestion, we provide per-video performance analysis on these three videos (“Honeybee,” “Jockey,” and “ShakeNDry”), and give additional experimental results on other temporally dynamic videos; please refer to our [Q3] below.
>
>
> ---
>
> **[W4] Lack of comparison with learning-based video compression methods.**
>
> As we mentioned earlier (in [W1] and [W2]), our main focus is not on video compression and
> we do not compare with state-of-the-art video compression methods. However, we think investigating our work further for compression is worth it, as our approach has the potential to mitigate the limitations of learning-based compression schemes. Recall that most learning-based video compression methods learn the compression procedure from video “datasets” (e.g., [1]), while our method requires and uses only a “single test video” for representing a video as succinct parameters. Hence, while existing methods mostly suffer from the distributional shift [2], our approach does not, since it works in a “zero-shot” manner. Not limited to, compared with traditional codecs, our method can provide other intriguing properties that the existing codecs cannot achieve, e.g., video inpainting, video frame interpolation, super-resolution, etc. We think finding a better method of video compression based on our work should be definitely an interesting future direction.
>
> ---

---

### Author Response · Authors · 2022-08-02
**General Response**

Dear reviewers and AC,

We deeply appreciate your efforts and time in reviewing our manuscript.

Our work proposes a novel coordinate-based neural representation (CNR) for videos, coined NVP, based on proposing learnable positional features that effectively amortize a given video as latent codes. As highlighted by reviewers, our paper is well-written (kkNU, MBhC), well-motivated to tackle the efficiency trilemma of video CNR (kkNU, MBhC), presents a new idea to learn the keyframes (LDYR, kkNU), while showing a solid empirical result (kkNU, MBhC).

We appreciate the reviewers’ insightful, incisive comments on our paper. To answer your concerns and questions, we have updated the manuscript and our project page (https://neurips2022-nvp.github.io) with the following additional experiments and clarification:

- Showing more compelling properties (video inpainting and video super-resolution) of NVP (Section 4.2) other than video frame interpolation and video compression
- Validation of our method on videos with larger temporal variations (Appendix F, project page)
- Additional experiment on a larger video (high-resolution, long) to show the scalability of our method (Appendix F, project page)
- Additional experiment on a Big Buck Bunny video, following the prior setup in NeRV [1] (Appendix F)
- Revision of Abstract and Introduction to clarify the focus of our work (Section 1 and 2)
- More extensive ablation studies to validate each component of NVP (Table 3)
- Video-wise compression results in the UVG benchmark (Appendix G)

We temporarily highlighted these updates as "blue" for your convenience.

Thank you very much !
Authors

---
[1] Chen et al., NeRV: Neural Representations for Videos, NeurIPS 2021.

---

> ### Author Response · Authors · 2022-08-09
> **Update on Revision (#1)**
>
> Dear reviewers and AC,
>
> Thank you for your time and efforts in reviewing our manuscript.
>
> This letter is to notice that we have made an additional update to the revision, which further incorporates the comments from Reviewer kkNU during the discussion period:
>
> - Adjust boldfacing of the ablation study result (Table 3)
> - Additional ablation study on the role of each component in its early training stage (Table 3)
> - Clarification of the role of modulated implicit function (Section 3.1, Section 4.3)
>
> These updates are temporarily highlighted in “red” upon the previous revision.
>
> If you have time, please check out the updated manuscript and the individual responses, and let us know if there are any additional concerns or questions. We will be happy to respond to your further comments during the remainder of the author discussion period.
>
> Thanks,
> Authors

---

### Author Response · Authors · 2022-08-05
**A gentle reminder**

Dear Reviewers,

Thank you for your time and efforts in reviewing our paper.

We believe that we sincerely and successfully address your concerns and questions, with the results of the supporting experiments. We also believe that our paper becomes much stronger through the clarification.

If you have any further concerns or questions, please do not hesitate to let us know and we will be happy to get back to you to clarify them.

Thank you very much!

Authors

---

> ### Comment · Reviewer_kkNU · 2022-08-08
> **Post rebuttal**
>
> I'd like to thank the authors for the rebuttal. The rebuttal clarifies many of my concerns and I think this is a technically solid work with good validation thus, I'm happy to keep my initial recommendation.
>
> One comment re boldfacing: when boldfacing one should take into account std estimates and not only look at the means. For example, when looking at the ablation study with std values one could clearly see that the benefit of modulation is not statistically significant and the last two lines in the table should be boldfaced. This suggest that the importance coming from the modulation might not be as large as suggested by the current text. Please correct the boldfacing accordingly and adjust the text accordingly if the paper gets accepted.

---

> > ### Author Response · Authors · 2022-08-09
> > **Response to Reviewer kkNU**
> >
> > Thank you for your response and for providing additional feedback on our manuscript! We decided to further address your comments with the additional experiments and the revision, which are temporarily marked as “red” for your convenience.
> >
> > As you pointed out, the effect of modulation is often marginal in its final training stage. In this respect, we fully agree that boldfacing the last two lines in Table 3 would be more clear for readers, and we updated Table 3 accordingly following your suggestion.
> >
> > Nevertheless, we do remark that the modulated implicit function is beneficial to achieving high-quality encoding within a small number of training steps, e.g., necessary for real-time or mobile applications. To validate this, we conducted additional experiments to highlight the role of modulated implicit function at early training iterations. As shown in the below table, similar to the other two components of NVP, modulated implicit function improves encoding quality measured in PSNR metric (higher is better) from 32.15 to 34.85 compared with a naive multilayer perceptron after 1,500 iterations ($\approx$7 minutes). We updated Section 3.1, Section 4.3, and Table 3 to clarify and add the importance/role of the modulated implicit function. Thank you very much again and we believe that this further strengthens our paper.
> >
> >
> > \begin{array}{c c c c c}
> > \hline
> > \text{Keyframes} & \text{Sparse feat.} & \text{Module.}  & \text{\\# Params.} & \text{PSNR (1.5K)} & \text{PSNR (150K)} \newline
> > \hline
> > \color{red}\tt{X} & \color{green}\checkmark & \color{green}\checkmark & \text{136M} &29.95\small{\pm2.69} & 31.21\small{\pm2.80} \newline
> > \color{green}\checkmark & \color{red}\tt{X} &  \color{green}\checkmark & \text{138M} &29.88\small{\pm4.99} & 32.44\small{\pm4.48}\newline
> > \color{green}\checkmark &  \color{green}\checkmark & \color{red}\tt{X} & \text{147M} &32.15\small{\pm3.08} & \mathbf{38.04\small{\pm2.27}}\newline
> > \hline
> > \color{green}\checkmark &  \color{green}\checkmark & \color{green}\checkmark & \text{136M} & \mathbf{34.85\small{\pm2.69}} & \mathbf{38.89\small{\pm2.11}}\newline
> > \hline
> > \end{array}

---

### Meta-Review · Area_Chair_g69S · 2022-08-26

**Recommendation:** Accept
**Confidence:** Less certain

**Metareview:**

The paper proposes a coordinate-based architecture for representing a video in a parameter-efficient and computationally efficient manner. Such architecture can be used for video compression, inpainting and frame interpolation.

The initial concerns were addressed during the rebuttal period and all the reviewers had a positive opinion of the paper. I therefore recommend acceptance.

**Award:**

No

---

### Decision · Program_Chairs · 2022-09-14

Accept